# Evolution of the electronic structure in open-shell donor-acceptor organic semiconductors

Zhongxin Chen [1], Wenqiang Li [1], Md Abdus Sabuj[2], Yuan Li [1✉], Weiya Zhu[1], Miao Zeng[1], Chandra S. Sarap [2], Md Masrul Huda[2], Xianfeng Qiao[1], Xiaobin Peng[1], Dongge Ma[1], Yuguang Ma[1], Neeraj Rai[2✉] & Fei Huang [1✉]

Most organic semiconductors have closed-shell electronic structures, however, studies have revealed open-shell character emanating from design paradigms such as narrowing the bandgap and controlling the quinoidal-aromatic resonance of the π-system. A fundamental challenge is understanding and identifying the molecular and electronic basis for the transition from a closed- to open-shell electronic structure and connecting the physicochemical properties with (opto) electronic functionality. Here, we report donor-acceptor organic semiconductors comprised of diketopyrrolopyrrole and naphthobisthiadiazole acceptors and various electron-rich donors commonly utilized in constructing high-performance organic semiconductors. Nuclear magnetic resonance, electron spin resonance, magnetic susceptibility measurements, single-crystal X-ray studies, and computational investigations connect the bandgap, π-extension, structural, and electronic features with the emergence of various degrees of diradical character. This work systematically demonstrates the widespread diradical character in the classical donor-acceptor organic semiconductors and provides distinctive insights into their ground state structure-property relationship.

[1] Institute of Polymer Optoelectronic Materials and Devices, State Key Laboratory of Luminescent Materials and Devices, South China University of Technology, Guangzhou 510640, P. R. China. [2] Dave C. Swalm School of Chemical Engineering and Center for Advanced Vehicular Systems, Mississippi State University, Mississippi State, MS 39762, United States. ✉email: celiy@scut.edu.cn; neerajrai@che.msstate.edu; msfhuang@scut.edu.cn

The last few decades have witnessed the rise of narrow bandgap organic semiconductors (OSCs) based on small molecules and polymers[1–3]. These materials have afforded a generation of optoelectronic technologies owing to their synthetic modularity, ease of manufacture, a wide array of desirable optical and transport properties, and opportunities for innovation in device structure not possible using inorganic materials[4,5]. OSCs with progressively narrower bandgaps are technologically relevant for many emerging optoelectronic applications, including harvesting low-energy photons in organic photovoltaics (OPVs)[6], infrared organic photodetectors (OPDs)[7,8], organic field-effect transistors (OFETs)[9], thermoelectrics, etc[10]. The successful engineering of these materials has relied on utilizing electron-rich donors (D) and electron-deficient acceptors (A) to construct D-A materials with strong intramolecular charge-transfer (ICT) interactions. While these materials exhibit improved efficiencies for charge separation and transport, high chemical stability, and promising performance in OPVs, OPDs, OFETs, and other emerging technologies, there are still many complexities associated with developing a cohesive relationship between electronic structure and performance. A fundamental challenge limiting the rational design of these materials is to understand the molecular, structural, and electronic factors that enable a transition from a closed- to open-shell electronic structure and connect these features with physicochemical properties and (opto)electronic functionality.

Unlike the conventional OSCs that display a strong electron pairing, open-shell diradicals exhibit weaker intramolecular electron-electron coupling. This diminished covalency is described by the diradical character index ($y_0$) and is closely related observable, the singlet-triplet energy gap ($\Delta E_{ST}$)[11,12]. This class of molecules displays unique singlet fission[13], non-linear optical[14], spin[15], magnetic[16,17], and multifunctional phenomena[12] promising for next-generation nanoelectronic and optoelectronic technologies[18]. Various open-shell diradicals based on archetypal π-conjugated molecules have been developed, including numerous analogs of Tschitschibabin's hydrocarbons (Fig. 1a)[19], extended para-quinodimethanes[20,21], quinoidal oligothiophenes[22], indenofluorenes[23], and other diverse polycyclic aromatic hydrocarbons (PAHs) (Supplementary Fig. 1)[11,21,24].

In general, design guidelines rely on the aromatization of embedded quinoidal subunits, which diminish the covalency of a π-bond and result in stabilization of the biradicaloid resonance form, a manifestation of Clar's π-sextet rule[20,25]. π-extended variants of Kekulé diradicals such as fused phenalenyl-based hydrocarbon reported by Kubo et al. exhibits higher $y_0$ and narrower bandgap as a function of size (Fig. 1a)[26]. Wang et al. provided a facile strategy to stabilize Tschitschibabin's hydrocarbons by replacing the carbinyl centers with isoelectronic aminium centers[27]. Steric protection of spin centers has enabled air- and temperature-stable diradicaloids such as diindeno[b, i]anthracene and zethrene frameworks (Fig. 1a)[23,28]. The recent report on the diradicaloid molecular cage revealed that the open-shell singlet ground state and global aromaticity were determined by the structural symmetry and the number of delocalized π electrons[29]. A narrowing of the bandgap in π-extended systems leads to increased admixing of the frontier molecular orbitals (FMOs) in their ground state, increasing $y_0$[24]. However, the synthesis of these π-extended open-shell species using traditional wet chemistry approaches continues to remain challenging, and the relatively poor stability limits their practical applications as functional materials[16,30].

The PAHs mentioned above with embedded quinoidal subunits, extensive π-delocalization, and narrower bandgaps have demonstrated diradical character in their neutral forms. They have been extensively studied in the past more than 110 years[19–30]. In our previous work, we reported the universal electron spin resonance (ESR) signals of the classical D-A narrow bandgap OSCs, and disclosed that the intrinsic open-shell quinoid-radical resonance structure act as the origin of their ESR signal (Fig. 1b)[31]. After we published our work, we found that Bhanuprakash and Wudl et al. have reported the open-shell character of the narrow bandgap D-A small molecules and polymers based on benzo[1,2-c;4,5-c]bis[1,2,5]thiadiazole (BBT) in 2011 and 2015, respectively (Fig. 1c)[32,33], and we regret this oversight on our part. Wu et al.[34,35] demonstrated that BBT-based oligomers showed pronounced diradical character and a reduced $\Delta E_{ST}$ as a function of chain length (Fig. 1c) with a combination of experimental and theoretical evidence. A recent report from Azoulay et al.[36] has also shown that the conjugated polymers comprising cyclopentadithiophene–thiadiazoloquinoxaline framework with very narrow bandgaps (0.5 eV < $E_g$ < 1.0 eV) exhibit high-spin ground state emanating from a high degree of electronic coherence along the π-conjugated backbone (Fig. 1c). Tam et al.[37] reported the proquinoidal-conjugated polymers based on 2,6-dialkyl-benzobistriazole, showing biradical character and high electrical conductivity. These results demonstrate that both Kekulé and non-Kekulé D-A open-shell systems have in common a series of electronic properties that give rise to diradical character, such as an increase in configuration mixing with a narrowing of the bandgap, extended conjugation, spatial distribution of the FMOs, etc[11,20,24]. In general, the nature of the ground state electronic structure of narrow bandgap D-A OSCs is often overlooked, and alternative lines of thinking and previous reports have attributed the presence of radicals and the associated ESR signals to the trapped oxygen[38], impurities, defects[39], polarons[40], or other species[41].

A clear and in-depth understanding of structure-property-electronic topology relationships is needed to reveal the origin of diradical character and the nature of electronic structure in narrow bandgap D-A OSCs. This contribution demonstrates that diradical character is widespread in narrow bandgap D-A materials, i.e., the open-shell singlet diradical structure appears in the canonical form (Fig. 1b). Diketopyrrolopyrrole (2,5-dihydropyrrolo[3,4-c]pyrrole-1,4-dione, DPP)[42,43], naphtho[1,2-c:5,6-c]bis([1,2,5]thiadiazole) (NT)[44,45], 2,1,3-benzothiadiazole (BT) and BBT cores were utilized in the construction of a series of small molecules in order to serve as representative examples of high-performance OSCs with progressively narrower bandgaps. In contrast to polymers, these small molecules have well-defined structures and eliminate complications associated with chain length, heterogeneity, purity, and disorder as a dominant influence when observing the development of the electronic structure across various molecular configurations. Nuclear magnetic resonance (NMR), ESR, superconducting quantum interference device (SQUID) magnetometry, single-crystal X-ray diffraction (XRD) studies, and electronic structure calculations clearly articulate an evolution of the electronic structure, as well as a prominent paramagnetic activity under practical operating conditions. These investigations provide an in-depth understanding of the structure-property-electronic-topology relationship in narrow bandgap OSCs.

## Results

**Optical properties and structural geometric parameters.** We have designed and synthesized a series of donor-acceptor-donor (D-A-D) small molecules comprised of thiophene flanked 2,5-bis(2-ethylhexyl)−3,6-di(thiophen-2-yl)−2,5-dihydropyrrolo [3,4-c]pyrrole-1,4-dione (TDPP) and NT acceptors with variations of the donor moieties (Fig. 2 and Supplementary Fig. 2). All compounds were synthesized as previously reported[31] or using standard Suzuki or Stille coupling protocols from the corresponding dibromo-substituted precursors with full details in the Supplementary Information. For the molecules based on the TDPP core, we

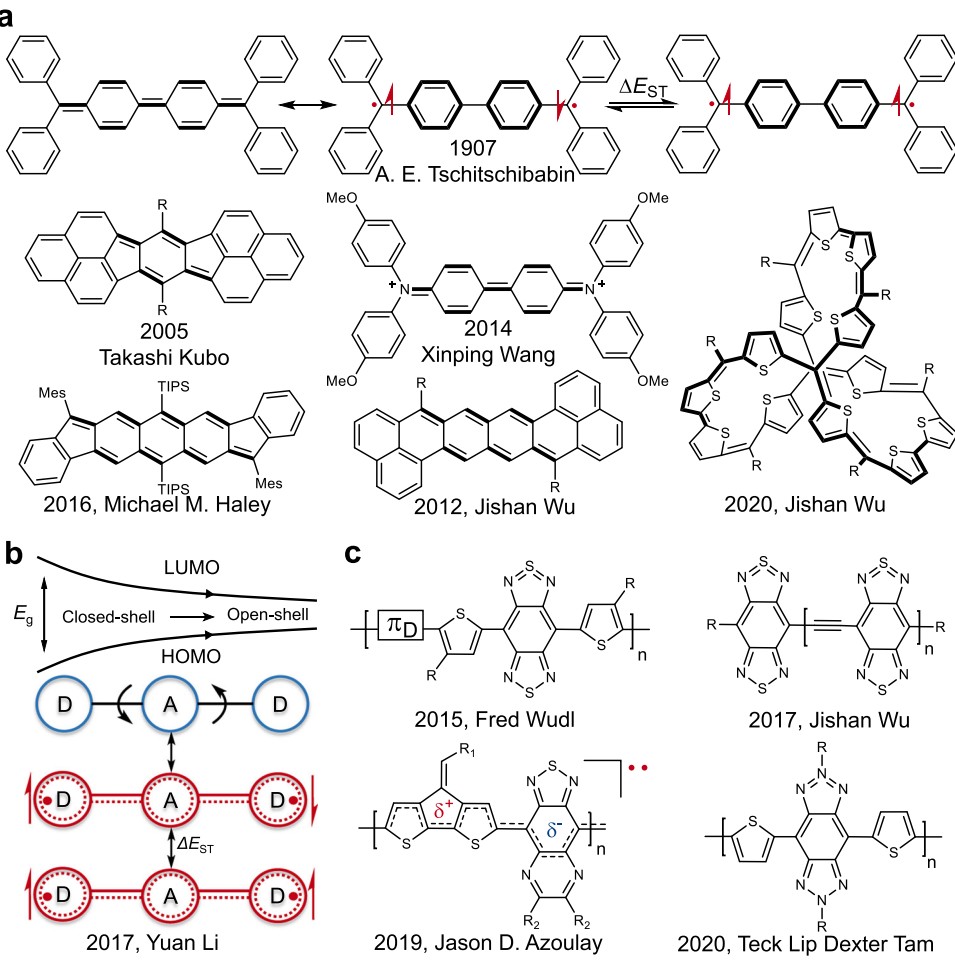

**Fig. 1 Molecules and polymers with open-shell ground states. a** Electronic structure of open-shell Kekulé-type Tschitschibabin's hydrocarbon with the resonance structure between the closed- and open-shell structures as well as the thermally excited triplet state[19]. Examples of typical compounds with a similar mechanism drawn in closed-shell resonance form[23,26–29]. **b** The widespread cross-over from closed- to open-shell character through narrowing of the bandgap of the donor-acceptor molecules initially reported in our previous work[31]. **c** Examples of donor-acceptor small molecules and polymers containing benzo[1,2-c;4,5-c]bis[1,2,5]thiadiazole and other similar acceptor units with open-shell character[32–36].

examined donors consisting of phenyl (Ph), 1-naphthyl (1 N), 2-naphthyl (2 N), anthracenyl (An), 1-pyrenyl (Py), thienyl (Th), fluorenyl (Flu), and 4-methoxy-*N*-(4-methoxyphenyl)-*N*-phenyla-nilinyl (TPAOMe) units as shown in Fig. 2a. For the molecules based on the NT core, thiophene and 4,4-didodecyl-4*H*-cyclo-penta[2,1-*b*:3,4-*b'*]dithiophene (CPDT) π-spacers were combined with the NT unit to provide 5,10-bis(4-hexylthiophen-2-yl) naphtho[1,2-*c*:5,6-*c'*]bis([1,2,5]thiadiazole) (NTT) and 5,10-bis(4,4-didodecyl-4*H*-cyclopenta[2,1-*b*:3,4-*b'*]dithiophen-2-yl)naphtho[1,2-*c*:5,6-*c'*]bis([1,2,5]thiadiazole) (NTC) (Fig. 2b), respectively. NTT and NTC were further π-extended with 2 N and Flu donors to furnish 2N-NTT, Flu-NTT, 2N-NTC, and Flu-NTC. These pro-quinoidal acceptor cores flanked by progressively electron-rich donor heterocycles induce narrowing of the bandgap across the series and increase the diradical character. The chemical structures and purity were confirmed by HPLC (Supplementary Fig. 3), NMR, elemental analysis, and mass spectra (Supplementary Information).

The optical properties were investigated using UV-vis-NIR absorption spectroscopy in chloroform (Supplementary Fig. 4) and as thin films (Fig. 2), and the electrochemical properties were studied through cyclic voltammetry (Supplementary Fig. 5). The optical bandgaps ($E_g^{opt}$) estimated from the intersection of absorption and emission curves of the pristine thin-films are shown below the chemical structures in Fig. 2 and listed in

Supplementary Table 1 together with the energy levels of the highest occupied molecular orbital (HOMO) and the lowest unoccupied molecular orbital (LUMO). The calculated wave-lengths of the first excited state using time-dependent density functional theory (TDDFT) at the PCM/BHandHLYP level of theory and 6-31 G(*d, p*) basis set (Supplementary Table 2) are in better agreement in the case of the NT-based molecules than the DPP-based materials. For the TDPP-based molecules, the An group exhibits a higher electron-donating ability than the Ph group. However, An-TDPP has a wider bandgap than Ph-TDPP ($E_g^{opt}$ = 2.02 versus 1.87 eV, respectively). Similarly, 1N-TDPP and 2N-TDPP incorporate a naphthyl unit differing only in the linking site between the D and A units; however, 2N-TDPP exhibits a significantly red-shifted absorption ($\lambda_{max}$ = 650 nm) and narrower bandgap ($E_g^{opt}$ = 1.84 eV) than 1N-TDPP ($\lambda_{max}$ = 600 nm, $E_g^{opt}$ = 1.96 eV). The difference can be explained by the steric repulsion between TDPP cores and conjugated groups. The steric repulsion induced by both the inner and outer ortho-hydrogens determines the extent of π-orbitals overlap. For example, two inner ortho-hydrogens of anthracene units of An-TDPP as shown in Supplementary Fig. 6, completely deviates from the plane of the thiophene flanked TDPP core with a value of 90° (see also Supplementary Figs. 7–9). Hence the orthogonal overlap of π-orbitals between the anthracene and thiophene units is significantly diminished, resulting in the restriction of

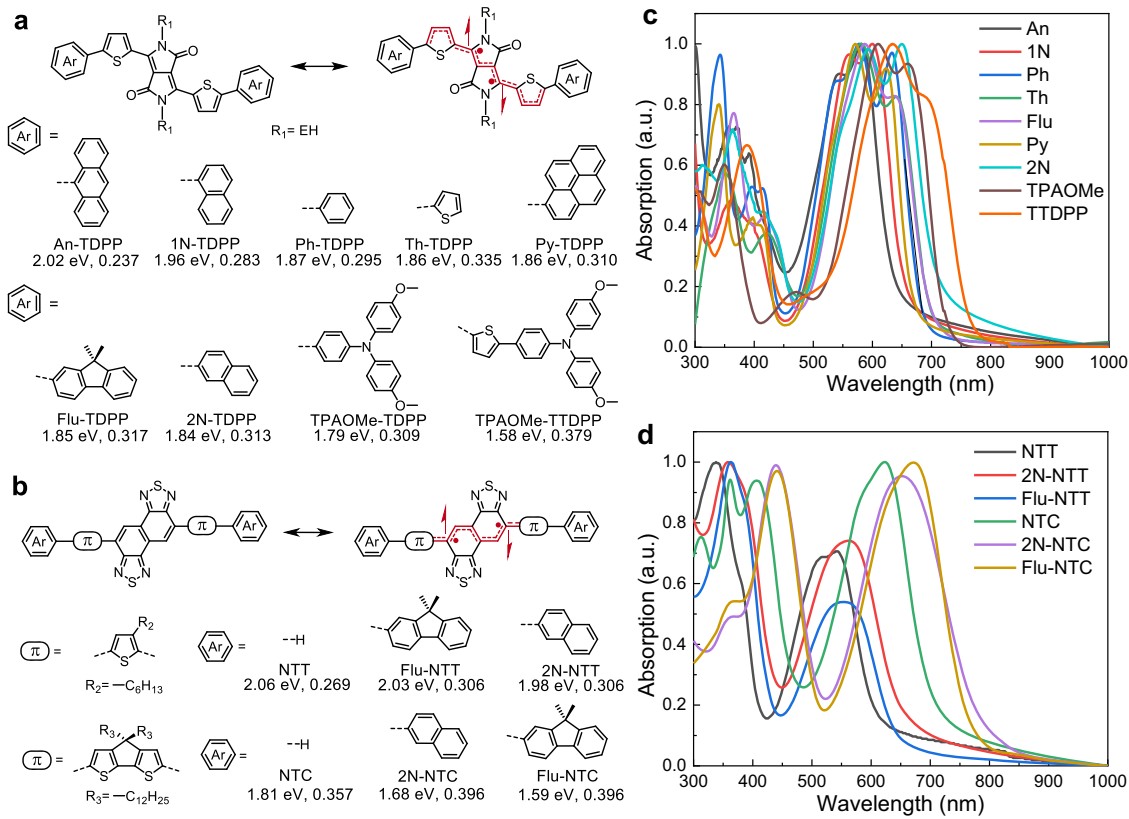

**Fig. 2 Molecular structures and thin-film absorption profiles of donor-acceptor-donor small molecules containing TDPP, NTT, and NTC cores.**
**a**, **b** Molecular structures of the DPP and NT-based small molecules with variations of the donor moieties. The optical bandgap ($E_g^{opt}$) as determined from the intersection of absorbance and emission curves of the thin-films and calculated $y_0$ at the PUHF/6-31 G($d,p$) level of theory, are listed beneath the donor units. **c**, **d** Absorption spectra of thin-films cast from chloroform onto quartz substrates.

π-delocalization from the former to the acceptor core. 1N-TDPP and Py-TDPP possess one outer and one inner ortho-hydrogens. Compared to the outer ortho-hydrogen, the inner one significantly decreases the overlap of π-orbital interactions due to the pronounced steric repulsion with the β-hydrogen of thiophene. As a result, the dihedral angle decreases to ~45° for 1N-TDPP and Py-TDPP derivatives, but the absorption pattern mainly depends on the π-extension (Py»1 N). Two outer ortho hydrogens of Ph and 2N-TDPP with a smaller dihedral angle of 25°, pushes the absorption to a longer wavelength and a greater π-delocalization of 2 N leads to a further bathochromic shift. Torsional effects reduce the overlap of the frontier MOs between adjacent D and A moieties and localize them on the TDPP core (Supplementary Fig. 11), which reduces π-conjugation and increases the bandgap (Supplementary Table 1). TPAOMe acts as the strongest donor, followed by Flu, Ph, fused-phenyl (1 N, 2 N), and Th groups. The concomitant decrease of the bandgap can be associated with an increase of the electron-donating capability of the donor and extension of the conjugation as demonstrated by progressively red-shifted absorption for Th-TDPP, Flu-TDPP, TPAOMe-TDPP, and TPAOMe-TTDPP (Fig. 2a).

The density difference between the first singlet excited state ($S_1$) and ground state ($S_0$), as determined using TDDFT, also indicates the effect of ICT within these molecules on their optical properties (Supplementary Figs. 17–19). For An-TDPP and 1N-TDPP, an increase in the calculated dihedral angle ($\phi_\alpha = 90°$ versus 46°, respectively) results in the restriction of ICT within the core of the molecule (Supplementary Fig. 7). In contrast, an increase in planarity and conjugation in Th-TDPP, Flu-TDPP,

TPAOMe-TDPP, and TPAOMe-TTDPP leads to better charge transfer from the terminal D to A units. NTC-based derivatives (NTC, 2N-NTC, and Flu-NTC) exhibit smaller bandgaps when compared with NTT-based derivatives (NTT, 2N-NTT, and Flu-NTT), as demonstrated by red-shifted absorption profiles (Fig. 2b, Supplementary Fig. 4c, Supplementary Table 1, and Supplementary Table 2). This is due to the extended conjugation and enhanced quinoidal stabilization endowed by the CPDT unit. 3-Alkyl chains of thiophene in NTT derivatives disrupt the planarity leading to the increased bandgap. The planar structures and extended conjugation promoted by flanking the CPDT unit of NTC-based derivatives promote better electronic coherence along the conjugated backbone and stronger ICT character. The narrower bandgap of NTC derivatives agrees with the smaller dihedral angles, larger overlap of the frontier MOs, and more efficient ICT in agreement with molecular orbital diagrams, geometric structures, and density difference plots (Supplementary Table 2, Supplementary Fig. 8, Supplementary Fig. 12, and Supplementary Fig. 18).

**Diradical character and ground-state electronic structure.** The narrow bandgaps and D-A character motivated our investigation of the ground state electronic structure. ESR measurements were conducted on powders of all the materials using the same molar quantity. All the materials showed ESR signals with $g$-factors ($g$) ranging from 2.0027–2.0037 (Fig. 3a, b, Supplementary Fig. 20 and Supplementary Fig. 21), indicating unpaired or weakly paired carbon-centered radicals[46,47]. An increase in ESR signal intensity as the bandgap is narrowed is evident across both the DPP- and NT-based materials series (Fig. 3a, b). The zero-field splitting and

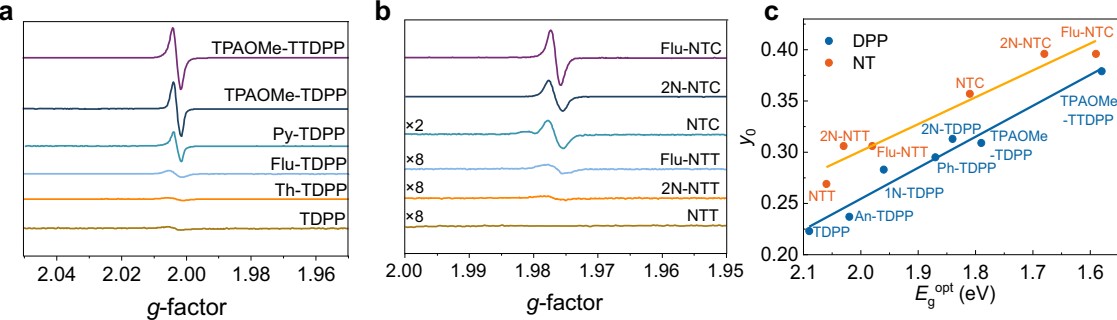

**Fig. 3 ESR spectra of TDPP, NTC, and NTT derivatives and diradical index as a function of bandgap. a, b** ESR spectra of powder samples comparing signal intensity using the same molar quantity of each material and under the same test conditions. **c** Calculated $y_0$ is plotted against experimental $E_g^{opt}$ obtained from the thin-film absorption profiles.

$\Delta m_s = \pm 2$ transitions of all the compounds were not detected, consistent with previous reports for diradicaloids[12]. Using the monoradical DPPH ($S = 1/2$) as the standard, the spin concentration of TPAOMe-TTDPP and 2N-NTC were calculated as 0.15% and 0.17% $N_A$, respectively. The spin concentration of TPAOMe-TTDPP was similar to the typical $p$-QDM diradicaloid CN-TDPP (Supplementary Fig. 21), indicating the moderate diradical character of the D-A compound TPAOMe-TTDPP. For TPAOMe-TDPP, TPAOMe-TTDPP, NTC, and 2N-NTC, the ESR response weakens from powders to solutions, due to the solvent effect and diluted spin density in solution (Supplementary Fig. 22). High purity samples prepared using high vacuum sublimation in an inert environment showed identical properties consistent with the intrinsic diradical character rather than impurities or defects (Supplementary Fig. 23).

Theoretical calculations at the spin-projected unrestricted Hartree–Fock (PUHF) level of theory and 6-31 G($d$, $p$) basis set were carried out to further confirm the ground state electronic structure of these small molecules (Supplementary Table 2) (see the Supplementary Information for other methods used to compare the open-shell characters). The radical character is defined by the indexes, $y_0$ (diradical character index) and $y_1$ (tetraradical character index), where $y_i$ ($i = 0, 1$) can have values as $y_i = 0$ (closed-shell), $0 < y_i < 1$ (intermediate open-shell), and $y_i = 1$ (pure open-shell). All the molecules demonstrate a diradical character with narrower bandgaps leading to higher $y_0$ (Fig. 3c). In addition to the diradical index, we also report a closely related quantity called fractional occupation number weighted electron density ($N_{FOD}$) using finite temperature-density functional theory (FT-DFT), which quantifies the strong electron correlation, and a larger $N_{FOD}$ signifies multireference character[48].

Complementary spectroscopic verification of the ground state electronic structure was conducted through variable temperature (VT) $^1$H NMR spectroscopy. In general, the $^1$H NMR spectra of the materials with wider bandgaps exhibited relatively sharp peaks in the aromatic region at room temperature. In contrast, the spectra of narrower bandgap materials showed significant peak broadening, unresolved peaks, or a completely flat baseline (Supplementary Figs. 25–27). While TDPP exhibits sharp peaks between 8.5–9.5 ppm in CDCl$_3$ at room temperature, peak broadening becomes evident in moving from TDPP → Th-TDPP → Flu-TDPP. In contrast, unresolved peaks and a flat baseline are evident in moving from TPAOMe-TDPP to TPAOMe-TTDPP (Supplementary Fig. 26). For TPAOMe-TTDPP in CD$_2$Cl$_2$, broad peaks sharpen upon cooling from 303 to 243 K (Fig. 4a). Broadened $^1$H NMR signals that sharpen upon cooling are also evident in 2N-NTC (Fig. 4d); however, a further reduction in temperature resulted in strong aggregation

and poor solubility, precluding further investigation. Broadening of the $^1$H NMR signals with increasing temperature, a strong ESR response, and lack of a half field ESR triplet signal are typical features of diradicaloids with delocalized singlet open-shell ground states and result from the population of a thermally accessible triplet diradical form[21,24,25,30].

TPAOMe-TTDPP and 2N-NTC powders were further characterized using VT-ESR and SQUID measurements. For both samples, the ESR intensity increases with an increase in temperature (Fig. 4b, e), which is consistent with SQUID measurements, indicating a population of a paramagnetic state[21,23,28,30]. As shown in Fig. 4c, f, the product of molar magnetic susceptibility and temperature ($\chi_M T$) increased with temperature. A fit of the temperature-dependence to the Bleaney–Bowers equation results in a $\Delta E_{ST}$ of $-2275$ K ($-5.52$ kcal mol$^{-1}$) and $-2397$ K ($-4.76$ kcal mol$^{-1}$) for TPAOMe-TTDPP and 2N-NTC, respectively. Variable temperature and magnetic susceptibility measurements confirmed the open-shell singlet ground state electronic structure with large singlet-triplet energy splitting of these compounds, and the weak contribution of paramagnetic triplet species in ground state[20,23,29,49].

In the case of the DPP-based molecules, TDPP has the lowest calculated diradical index, $y_0 = 0.22$ and $N_{FOD} = 1.104$ (Supplementary Table 2 and Supplementary Table 3) in the series, which can be attributed to the lack of extended conjugation and the larger bandgap (Supplementary Table 2). As the bandgap decreases, $y_0$ and $N_{FOD}$ progressively increase across the series. The largest value is observed for TPAOMe-TTDPP ($y_0 = 0.379$ and $N_{FOD} = 2.816$), consistent with its broadened $^1$H NMR spectrum and significant ESR signal intensity. Moreover, computational results for TPAOMe-TDPP and TPAOMe-TTDPP are consistent with tetraradical character ($y_1 = 0.056$ and 0.112, respectively), which increases upon π-extension with a thiophene unit. The presence of a small polyradical character is also validated with the FT-DFT, showing large $N_{FOD}$ values (see Supplementary Table 3 and Supplementary Table 4). An-TDPP ($y_0 = 0.237$), 1N-TDPP ($y_0 = 0.283$), and Ph-TDPP ($y_0 = 0.295$) exhibit smaller diradical character following in trend with the planarity of the molecule. The large dihedral angle ($\phi_\alpha = 86.6°$, calculated = 90°) (Fig. 5 and Supplementary Fig. 7) in An-TDPP reduces $y_0$ when compared with the more planar 1N-TDPP ($\phi_\alpha = 46$, calculated) and Ph-TDPP ($\phi_\alpha = 25°$, calculated) analogues. NTC-based molecules with narrower bandgaps show higher $y_0$ and $N_{FOD}$ values than the NTT-based molecules. In agreement with increased planarity and extended conjugation, the NTC molecules develop increasing $y_0$ in moving from NTC ($y_0 = 0.357$) to Flu-NTC and 2N-NTC ($y_0 = 0.396$). NTT ($y_0 = 0.269$), 2N-NTT ($y_0 = 0.306$), and Flu-NTT ($y_0 = 0.306$) exhibit lower diradical character. We also investigated the

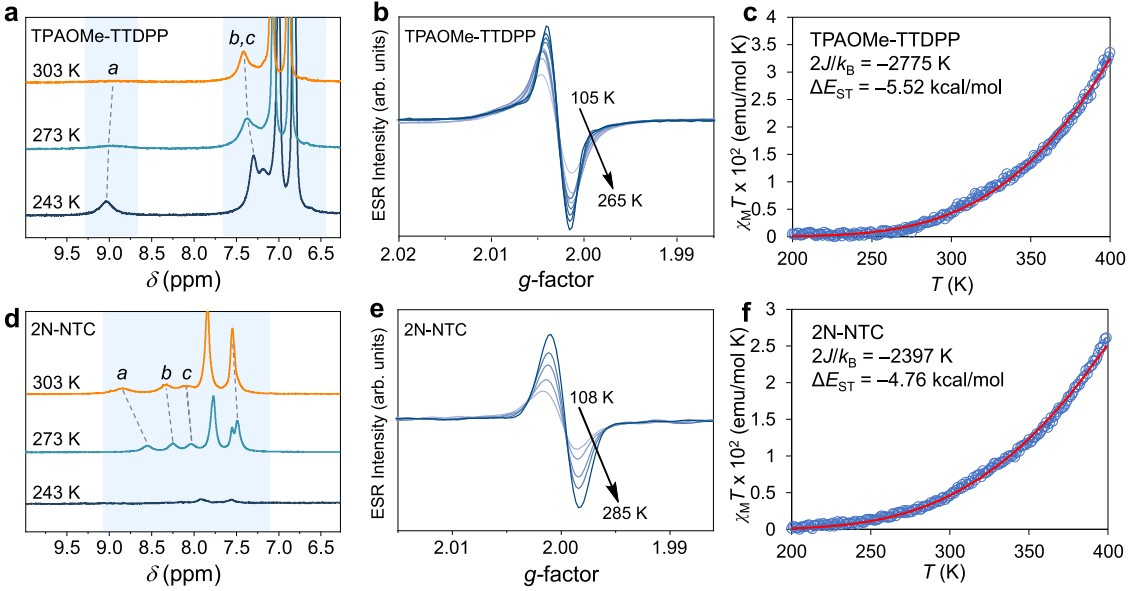

**Fig. 4 Temperature dependent properties of TPAOMe-TTDPP and 2N-NTC. a, d** Stacked VT $^1$H NMR spectra in $CD_2Cl_2$. **b** VT ESR spectra measured from 152–284 K with $g = 2.0031$ and (**e**) from 105–285 K with $g = 2.0029$. **c, f** SQUID magnetometry of the solid sample showing magnetic susceptibility times temperature $\chi_M T$ vs. $T$ from 200–400 K, fit to the Bleaney−Bowers equation with $g = 2.003$, giving $\Delta E_{ST}$ ($2J/k_B$) (red line).

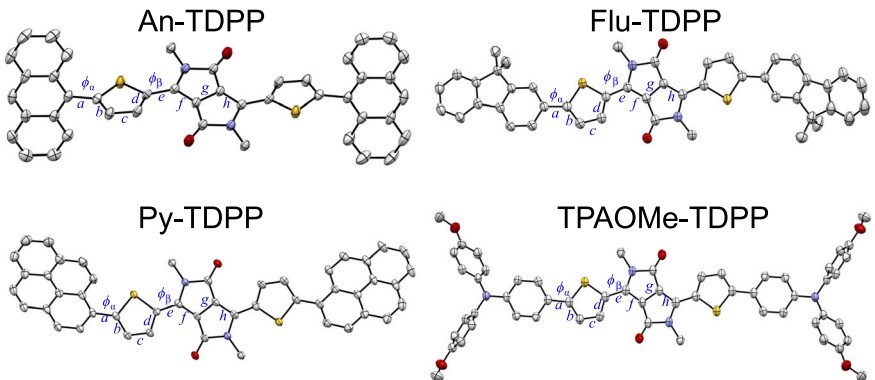

**Fig. 5 Solid-state structures by XRD.** ORTEP drawings with side-chains and hydrogen atoms omitted for clarity. Thermal ellipsoids are drawn at 50% probability. Selected bond lengths (Å) and dihedral angles ($\phi$) for An-TDPP, Flu-TDPP, Py-TDPP, and TPAOMe-TDPP, respectively: $\phi_\alpha$ (86.6°, 2.78°, 40.7°, 14.4°), $\phi_\beta$ (13.9°, 11.4°, 5.71°, 5.17°), $a$ (1.475, 1.464, 1.469, 1.461), $b$ (1.356, 1.369, 1.380, 1.367), $c$ (1.403, 1.400, 1.396, 1.397), $d$ (1.371, 1.372, 1.385, 1.376), $e$ (1.442, 1.443, 1.432, 1.437), $f$ (1.388, 1.384, 1.387, 1.404), $g$ (1.408, 1.408, 1.412, 1.406), $h$ (1.388, 1.386, 1.403, 1.404).

diradical character of the BT and BBT analogs flanked by TPAOMe units (Supplementary Fig. 2). The largest $y_0$ is obtained for TPAOMe-BBT ($y_0 = 0.665$, $N_{FOD} = 3.017$) followed by TPAOMe-NTT ($y_0 = 0.304$, $N_{FOD} = 2.636$) and TPAOMe-BTT ($y_0 = 0.264$, $N_{FOD} = 2.277$), which is consistent with the ESR signal intensity (Supplementary Fig. 21) and $E_g^{opt}$ (Supplementary Table 1) from TPAOMe-BTT to TPAOMe-NTT to TPAOMe-BBTT. The increase in diradical character across this series is attributed to the increased pro-quinoidal nature of the cores and the enhanced planarity and conjugation (Supplementary Fig. 9 and Supplementary Fig. 13)[33,36]. These results highlight the capability to effectively tune the bandgap and diradical character as a function of each donor-acceptor pair's structural and electronic characteristics, which ultimately controls their FMO interactions.

**X-ray crystallographic analysis and theoretical calculations**. We provide additional insight into the ground state structure through single-crystal XRD studies (Fig. 5) complemented by NICS$_{iso}$(1) (isotropic nucleus independent chemical shift at 1 Å above rings'

plane) calculations and ACID (anisotropy of the induced current density) plots of key structures (Fig. 6)[50,51]. Single crystals of An-TDPP, Py-TDPP, Flu-TDPP, and TPAOMe-TDPP were obtained by slow evaporation from either methanol or chloroform solutions (see Methods). The corresponding DFT optimized geometric parameters for the structures are compared in Supplementary Fig. 7 and Supplementary Table 6, and the solid-state packing of these crystals are depicted in Supplementary Figs. 28-31. The dihedral angles $\phi_\alpha$ and $\phi_\beta$ in An-TDPP are 86.6° and 13.9°, respectively, which are both the largest of the series (Fig. 5). The non-planar backbone of An-TDPP inhibits in-plane delocalization, localizing spins closer to the DPP core consistent with $g = 2.0034$. In contrast, Py-TDPP, Flu-TDPP, and TPAOMe-TDPP exhibit smaller dihedral angles, promoting more extended conjugation as indicated by their more delocalized frontier MOs (Supplementary Fig. 11).

Figure 5 further compares the bond lengths connecting the end-capping unit ($a$, $m$) to the thiophene, across the thiophene ($b$, $c$, and $d$), from the thiophene to the core ($e$, $i$), and across the DPP core ($f$, $g$, and $h$). Significant bond length alternation (BLA)

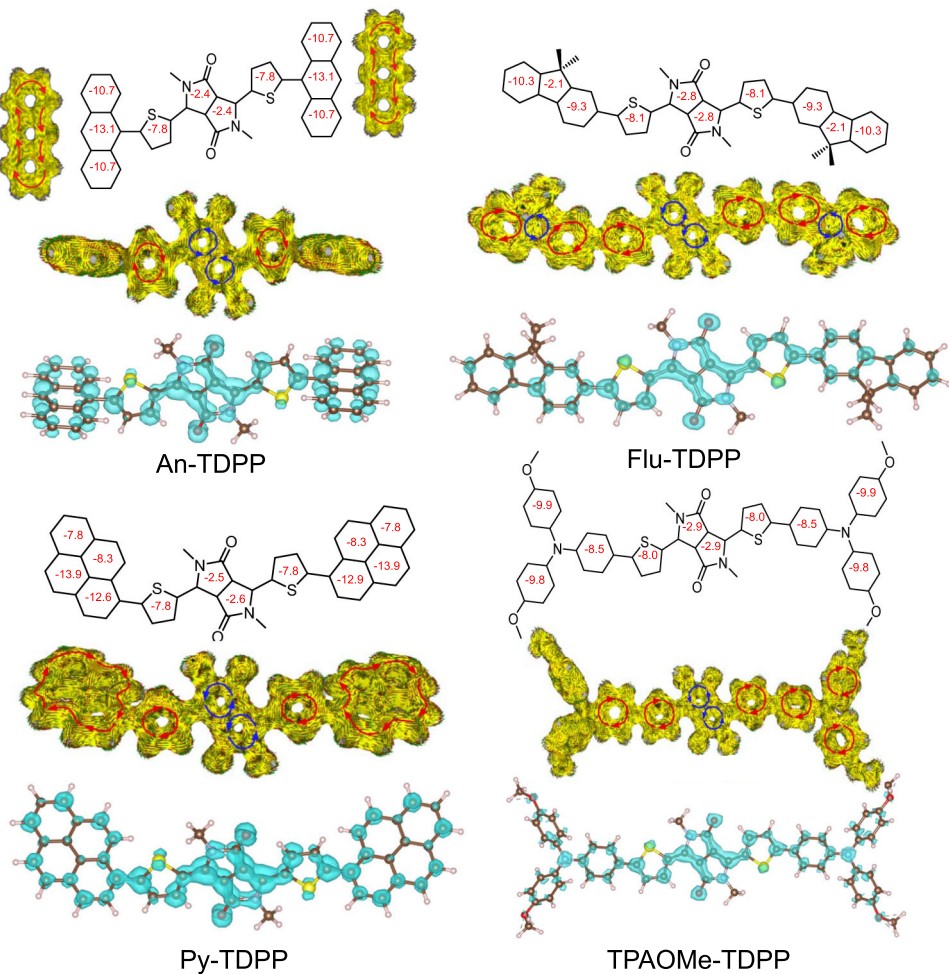

**Fig. 6 Calculated NICS$_{iso}$(1) values, ACID, and FOD plots for An-TDPP, Flu-TDPP, Py-TDPP, and TPAOMe-TDPP.** NICS$_{iso}$(1) and ACID are calculated at RB3LYP/6-31 G(d,p) level of theory and basis set. The red and blue arrows indicate clockwise (diatropic: aromatic) and counterclockwise (paratropic: quinoidal) ring current, respectively. The applied magnetic field is perpendicular to the molecular backbone and points out through the plane of the paper. ACID plots were generated with an isovalue = 0.025 au. FOD plots (σ = 0.002 e/Bohr[3]) are obtained from the FT-DFT at B3LYP/6-31 G(d,p) level of theory and basis set.

is observed, spanning lengths from 1.356–1.475 Å. Overall, a decrease in the absolute value of the BLA is evident in going from An-, Py-, Flu-, to TPAOMe-TDPP, consistent with the comparatively smaller bandgaps measured. The solid-state structures show a slight elongation in the length of *e* and *i* relative to the computed values for An- (1.442, calc. 1.440 Å), Flu- (1.443, calc. 1.436 Å), and TPAOMe-TDPP (1.437, calc. 1.435 Å), indicating greater single bond character in the crystal structure. However, Py-TDPP (1.432, calc. 1.437 Å) shows a contraction. The values for bonds *b*, *c*, and *d* suggest that the thiophene unit becomes more delocalized in moving from An- → Flu- → TPAOMe-TDPP as evidenced by a reduced BLA. The length of bond *g* decreases in moving from Py- → Flu- ~ An- → TPAOMe-TDPP, indicating increasing double-bond character towards the quinoidal form accompanied by a decrease in BLA in the core of the resonance hybrid. A similar trend in open-shell DPP-based materials has been reported showing the analogous bond *g* contracts, while *f* and *h* increase[52]. In general, a smaller but non-zero BLA of the linking thiophenes indicates a delocalized double-bond character[20]. Bond *e* shows an elongation in An-, Flu-, and TPAOMe-TDPP from the computed structure, while Py-TDPP shows a contraction. These structural features between the resonance hybrid and the computed structure suggest contributions from both the open-shell and the closed-shell

resonance forms at 100 K. In TPAOMe-TDPP, there is almost no BLA within the DPP core, suggesting bond-length equalization (*f* = 1.404 Å, *g* = 1.406 Å, *h* = 1.404 Å).

TPAOMe-TDPP shows the strongest ESR signal intensity consistent with the higher $y_0$ and enhanced delocalization. TPAOMe-TTDPP shows a further increase in delocalization accompanied by an increase in ESR intensity. For Py-TDPP, the NICS$_{iso}$(1) values for the thiophene and pyrene end groups are the most aromatic of the four molecules (Fig. 6 and Supplementary Fig. 7). BLA analysis reveals a higher degree of asymmetry within the core when compared to the other materials. Py-TDPP shows increased local aromaticity in the attached thiophene and ancillary donor units, diminishing delocalization from the core since the aromatic contribution disrupts adaptation of the quinoidal form. The combination of these features in Py-TDPP collectively inhibits delocalization through electronic effects, localizing spins on the DPP core and resulting in enhanced spin-orbit coupling. The localization of the unpaired spins is readily visible from the FOD plots (Fig. 6, Supplementary Figs. 14–16), where the density distribution is mostly observed in the materials core with a small contribution in the adjacent thiophene units.

Crystals of Flu-TDPP-C8 with linear octyl (-C$_8$H$_{17}$) and TPAOMe-TDPP-C4 with linear butyl (-C$_4$H$_9$) chains were

examined to understand the effect alkyl chain has on the structure, molecular planarity, and resultant properties. Steric perturbations result in deviations in the bond lengths and molecular configuration, resulting in significant changes to the electronic topology[30]. The dihedral angles of Flu-TDPP-C8 ($\phi_\alpha = 20.54°$, $\phi_\beta = 16.26°$) and TPAOMe-TDPP-C4 ($\phi_\alpha = 23.44°$, $\phi_\beta = 7.62°$) are comparatively larger than that of Flu-TDPP and TPAOMe-TDPP illustrating that a more planar backbone is adopted using 2-ethylhexyl (EH) side-chains. Flu-TDPP and TPAOMe-TDPP thus displayed an increased ESR response compared with Flu-TDPP-C8 and TPAOMe-TDPP-C4, respectively (Supplementary Fig. 24). Therefore, a planar backbone promotes better electronic coherence, increases delocalization, and increases diradical character. Moreover, Flu-TDPP and TPAOMe-TDPP exhibit larger π-π stacking distances of 3.642 Å and 3.645 Å, respectively compared to Flu-TDPP-C8 (3.518 Å) and TPAOMe-TDPP-C4 (3.291 Å) (Supplementary Table 6) associated with the increased steric bulk of the branched chains. This feature is known to influence spin-spin interactions in the solid-state.

$NICS_{iso}(1)$ calculations were further utilized to analyze the aromatic/quinoidal character of the individual rings along the π-framework, where a large negative $NICS_{iso}(1)$ values correspond to greater aromaticity of a particular ring. The $NICS_{iso}(1)$ values (Fig. 6, Supplementary Figs. 7–9) are more negative (aromatic) at the periphery of the molecules and less negative at the core. This generalization holds for all of the DPP and NT molecules examined. Moreover, the $NICS_{iso}(1)$ values of the central acceptor cores become more aromatic in moving from the singlet to the triplet state (Supplementary Fig. 10). ACID plots (Fig. 6) were generated to further confirm the aromatic and quinoidal character of the molecular backbone and show a strong diatropic ring current at the peripheral benzenoid rings and distinct paratropic current at the π-conjugated core consistent with $NICS_{iso}(1)$ calculations and BLA analysis of the ground state structures[49].

## Discussion

In summary, this work demonstrates the widespread existence of open-shell (diradical) character in alternating D-A molecules comprised of building blocks commonly utilized to construct narrow bandgap OSCs. In contrast to previous work on pro-aromatic or quinoidal Kekulé-type molecular systems, these materials show significant ICT character between electron-rich and electron-deficient components within the π-conjugated backbone. The construction of D-A materials afforded a systematic investigation of the effects of the bandgap, π-extension, ICT character, structural, and electronic features with the emergence of various degrees of diradical character. Temperature-dependent NMR, ESR, SQUID, and single-crystal XRD complemented by theoretical studies provide a clear articulation of how molecular, structural, and electronic features incrementally and rationally influence diradical character in these materials. A progression from $y_0 = 0.22 \rightarrow 0.67$ using these design guidelines demonstrates that stable materials with variable diradical character can be realized, circumventing demanding, multistep synthetic approaches traditionally utilized to access these materials. The capability to create different classes of open-shell diradicals from D-A building blocks opens opportunities to better understand the properties of D-A OSCs, and realize (opto)electronic functionalities leading to various technologies such as electronics, superconductivity, spintronics, and quantum devices[53–57].

## Methods

**Synthesis and characterization of the donor-acceptor molecules**. All the donor-acceptor based molecules were synthesized using standard Suzuki and Stille-coupling protocols as detailed in Supplementary Information. Chemical structures and purity

were confirmed by [1]H NMR and [13]C NMR spectroscopy using a Bruker Avance 400 MHz spectrometer in CDCl$_3$ at room temperature. HPLC analysis was conducted using the Cosmosil 5C$_{18}$-MS-II column to give the purity information.

**Optical absorption spectroscopy**. UV-Vis-NIR spectra were recorded from $350-1000$ nm using a Shimadzu UV-3600 spectrophotometer. Thin films were prepared by spin-coating a 10 mg mL$^{-1}$ chloroform solution onto a quartz substrate at 1500 rpm for 30 s.

**Electron spin resonance spectroscopy**. Continuous-wave X-band ESR spectra of powder samples were recorded on Bruker ELEXSYS-II E500 CW-ESR spectrometer. The molar quantity of the material was 0.02 mmol. The spectra were recorded at room temperature at a signal attenuation of 20 dB in quartz ESR standard quality tubes with an outer diameter of 5 mm from Beijing Synthware glass.

**Superconducting quantum interference device**. Magnetic measurements were carried out on a Quantum Design MPMS XL superconducting quantum interference device. The sample was prepared by placing 8.8 mg of 2N-NTC and 9.0 mg of TPAOMe-TTDPP in a polycarbonate capsule. Prior to measurement, the background of the empty capsule was recorded. Temperature-dependent data was collected from T = 2–400 K at an applied field of 5000 Oe. The susceptibility data were corrected for the diamagnetism of the sample holder and the intrinsic diamagnetism of the sample. The magnetic susceptibility of TPAOMe-TTDPP and 2N-NTC was fitted using the following equation:

$$\chi = \frac{2Ng^2\mu_\beta^2}{kT}\left[\frac{1}{3 + e^{-2J/kT}}\right] \quad (1)$$

**X-ray crystallography**. Crystals of An-TDPP, Py-TDPP, Flu-TDPP, TPAOMe-TDPP, Flu-TDPP-C8, and TPAOMe-TDPP-C4 were grown by slow evaporation of a mixed dichloromethane and methanol solution. The three crystals were mounted on a MiTeGen MicroLoop with Paratone oil under an N$_2$ stream (100 K) in a Bruker AXS APEX2 diffractometer equipped with a charge-coupled device (CCD) detector and a Cu Kα microfocus source, with Quazar optics. The structure was solved using the SHELXL program and refined using the SHELX software suite and Olex2 program. All non-H atoms were refined anisotropically and H atoms refined isotropically and constrained to ideal geometries.

**Density functional theory calculations**. The Gaussian 16 program package[58] was used for the geometry optimizations of the low bandgap OSCs using hybrid density functional, B3LYP[59] and 6-31 G(d, p) basis set[60]. The local minima of the optimized structures were verified with frequency calculations. All conditions of the geometry optimizations were used as defaults. For all the low bandgap OSCs, the side chains were truncated to -CH$_3$. The diradical ($y_0$) and tetraradical ($y_1$) character were calculated with the spin-projected unrestricted Hartree-Fock (PUHF) method using Yamaguchi's formula[61] at UHF/6-31 G($d$, $p$) level of theory and basis set:

$$y_i = 1 - \frac{2T_i}{1 + T_i^2}; \; T_i = \frac{n_{HONO-i} - n_{LUNO+i}}{2} \quad (2)$$

where, $i = 0$, 1; $T_i$ is the overlap between the two corresponding orbitals; $n_{HONO}$ and $n_{LUNO}$ are the occupation numbers of the highest occupied natural orbital and the lowest unoccupied natural orbital, respectively. Nucleus independent chemical shift ($NICS_{iso}(1)$) was computed at 1 Å above the rings plane with the gauge-independent atomic orbital (GIAO) method[50]. The optical properties were modeled with TDDFT at PCM/BHandHLYP/6-31 G(d,p) level of theory and basis set and chloroform was used as an implicit solvent[62]. ACID (anisotropy of the induced current density) plots were generated using the method developed by Herges et al.[50] FT-DFT calculations are performed using the ORCA[63] program package and the open-shell characters are reported directly from the LUNOs occupancies, without utilizing the Yamaguchi's formula (Supplementary Table 4). The fractional occupation number weighted electron density ($N_{FOD}$) is computed by the method developed by Grimme et al.[48] $N_{FOD}$ globally quantifies the strong electron correlation. Molecules with a large $N_{FOD}$ will have a multireference character[45].

## Data availability

Synthetic procedures and characterizations for all the new compounds, including the NMR, ESR, mass spectra, HPLC, UV-vis-NIR absorption, PL, and CV spectra, are provided in the Supplementary Information. The X-ray crystallographic coordinates for structures reported in this study have been deposited at the Cambridge Crystallographic Data Centre (CCDC). CCDC number 1882131 (TPAOMe-TDPP), 1919550 (Py-TDPP), 1919552 (An-TDPP), 1908981 (Flu-TDPP), 1908984 (Flu-TDPP-C8), 1908979 (TPAOMe-TDPP-C4). These data can be obtained free of charge from CCDC via www.ccdc.cam.ac.uk/data_request/cif. The computational methodology, codes used to generate data, and the optimized geometry of molecules has been provided in the supplementary information. All experimental data queries should be directed to Y.L.,

and all computational data queries should be directed to N.R., Y.L. and N.R. will respond to data requests within one week of receiving the request. There is no restriction on the non-commercial use of the data. Commercial use of the data will be handled per the institutional policies where the respective work was performed.

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

## Acknowledgements

The authors acknowledge the financial support of the Basic and Applied Basic Research Major Program of Guangdong Province (No. 2019B030302007), Innovation Research Group Project of Fund Committee (No. 51521002), National Key Research and Development Program of China (No. 2019YFA0705900) funded by MOST, Natural Science Foundation of China (51973063, 21733005, 91633301), and the Science and Technology Program of Guangzhou (No. 201707020019). M.A.S., C.S.S., M.M.H., and N.R. acknowledge the financial support from the National Science Foundation (OIA-1757220) for the computational aspects of this project. This work used supercomputing resources at the high-performance computing center at Mississippi State University and the Extreme Science and Engineering Discovery Environment (XSEDE), which is supported by the National Science Foundation grant number ACI-1548562. This work used XSEDE Stampede 2 at the Texas Advanced Computing Center (TACC) through allocation TG-CHE140141. We thank Michael M. Haley in the University of Oregon and Juan Casado in the University of Malaga for helpful discussions. Jason D. Azoulay and Kevin S. Mayer (University of Southern Mississippi) provided the SQUID data and performed BB fit to get the ST gap, and technically edited the initial draft of the manuscript.

## Author contributions

Z.C., W.L., and M.A.S. contributed equally to this work. Y.L. proposed and conducted the whole research. Z.C., W.L. W.Z, and M.Z. carried out the synthesis and characterization of the materials. Z.C., Y.L., M.A.S, and N.R. wrote the manuscript. W.Z., C.S.S., and F.H. helped on the writing and revision of the paper. M.A.S. performed the DFT calculations and the associated analysis. C.S.S performed calculations using the FT-DFT method and contributed to the analysis of the (poly)radical character. M.M.H. developed the analysis code for ACID/NICS plots and contributed to the analysis of aromaticity. N.R. supervised the computational aspects of the project. X.Q., X.P., D.M., and Y.M. and the other authors discussed the results and commented on the paper.

## Competing interests

The authors declare no competing interests.
