## [Peer Review File · Nature Communications]

Evolution of the Electronic Structure in Open-Shell Donor-Acceptor Organic SemiconductorsREVIEWER COMMENTS

Reviewer #1 (Remarks to the Author):

This paper deals with structure-property-electronic topology relationship of open-shell donor-acceptor organic semiconductors.

The paper was well organized by various analysis using NMR, ESR, MSM, X-ray, and computational investigation. However, The result is predictable. These are the contents that have naturally been considered in the design of molecular structures. The optical properties of what wavelengths are absorbed in molecular structure design are important when reflecting the desired electronic characteristics. This study has limitations in making only optical properties predictable. Therefore, I do not recommend this paper to publish in Nature Communications

Reviewer #2 (Remarks to the Author):

This study is a follow-up to earlier publications from the Azoulay group in which the open-shell character of low band gap polymers was explored. The present study moves from polymeric materials, of which there are several examples, to a series of molecularly defined materials comprising common conjugated building blocks. The development of materials showing diradical or open-shell character is an active area of research, as figure 1 in the paper highlights. Many of these examples contain relatively unstable extended acenes or quinoidal materials. The significant finding of this paper, in comparison to the earlier work, is the more ubiquitous presence of diradical character even for the commonly observed DPP and NT containing materials. The authors show, via a variety of measurements, that a trend is observed, largely matching that from polymer examples, in which lower band gap materials show larger diradical character. I think this is an interesting finding, and the paper will certainly generate interest in the community. I believe it is suitable for publication in this journal, but there are some issues to address first.

Given the large absorption tails, how is the absorption onset defined in the measurements? I would suggest they use a more precise method than currently used, for example the intersection of absorption and emission, given the importance of the band gap to their discussion.

Similarly the authors comment on the long absorption tail being characteristic of diradical character, but given that the measurements are not corrected for reflection and scattering, I don't think they can assert this. I strongly suggest they provide film measurements which have been corrected for these factors.

Why is all EPR data in the solid state only? What happens in solution measurements? Is diradical character observed? If not, why not. This should be addressed in the manuscript.

Given the importance of purity in these sensitive measurements, I find it surprising that no information on molecular purity is given except for NMR. Many of the mass spectra show peaks at different mass to that of the molecular ion. Given that all of these materials were made by Pd coupling, can the presence of homo-coupled defects or varying Pd levels be ruled out? Some information on purity (for example HPLC trace, melting points, elemental analysis) should be provided.

No info is given on how 2N-TDPP and 2N-NTT were sublimed.

List of point-to-point response of reviewers' comments

Reviewer 1 (Remarks to the Author):

This paper deals with structure-property-electronic topology relationship of open-shell donor-acceptor organic semiconductors. The paper was well organized by various analysis using NMR, ESR, MSM, X-ray, and computational investigation. However, the result is predictable. These are the contents that have naturally been considered in the design of molecular structures. The optical properties of what wavelengths are absorbed in molecular structure design are important when reflecting the desired electronic characteristics. This study has limitations in making only optical properties predictable. Therefore, I do not recommend this paper to publish in Nature Communications.

Response: We thank the reviewer for taking time to review our manuscript and providing very helpful comments. In our manuscript, we propose that open-shell diradical character is prevalent in narrow bandgap organic semiconductors (OSCs). Nuclear magnetic resonance, electron spin resonance, magnetic susceptibility measurements, single-crystal X-ray studies, and computational investigations demonstrated that the open-shell singlet diradical character was closely related to the structural and electronic features. More importantly, we provide a new insight to understand the structure-property-electronic topology relationship of narrow bandgap OSCs from the perspective of ground-state electronic structure, which is rarely reported before. We highlight that the diradical character not only determines the optical properties. Previous reports demonstrated the potential of open-shell radical materials such as triphenylmethyl analogues, quinoidal oligothiophenes, polycyclic aromatic hydrocarbons, and high-spin donor-acceptor polymers in NIR light-emitting diodes, organic field effect transistors, organic photovoltaics, organic magnets, spintronic devices, and high conductivity materials (*Nature* 2018, 563, 536–540; *Chem. Soc. Rev.*, 2012, 41, 5672–5686; *Nat. Chem.*, 2016, 8, 753–759; *J. Am. Chem. Soc.* 2017, 139, 18376–18385; *Chem. Soc. Rev.*, 2012, 41, 303–349; *Adv. Funct. Mater.* 2020, 30, 1909805). Using structure-property-electronic topology relationship, we can rationally design and control the electronic properties of narrow bandgap OSCs for next-generation electronic and spintronic devices. Therefore, we believe that the investigation of ground-state open-shell diradical character are important and promising for the development of narrow bandgap OSCs in various applications.

Reviewer 2 (Remarks to the Author):

This study is a follow-up to earlier publications from the Azoulay group in which the open-shell character of low band gap polymers was explored. The present study moves from polymeric materials, of which there are several examples, to a series of molecularly defined materials comprising common conjugated building blocks. The development of materials showing diradical or open-shell character is an active area of research, as figure 1 in the paper highlights. Many of these examples contain relatively unstable extended acenes or quinoidal materials. The significant finding of this paper, in comparison to the earlier work, is the more ubiquitous presence of diradical character even for the commonly observed DPP and NT containing materials. The authors show, via a variety of measurements, that a trend is

observed, largely matching that from polymer examples, in which lower band gap materials show larger diradical character. I think this is an interesting finding, and the paper will certainly generate interest in the community. I believe it is suitable for publication in this journal, but there are some issues to address first.

Response: We are encouraged by the reviewer's positive assessment and the meaningful comments to improve our paper. We have carefully revised the manuscript based on the suggestions from the reviewer. The detailed point-to-point response to each comment and the modifications of manuscript are listed as below.

Comment 1: Given the large absorption tails, how is the absorption onset defined in the measurements? I would suggest they use a more precise method than currently used, for example the intersection of absorption and emission, given the importance of the band gap to their discussion.

Response: Thank for the helpful comment. In our earlier version of the manuscript, we determined the absorption onset by the intersection of the linear fit trendline of the absorption spectrum and the tangent of absorption tail (*Adv. Mater.* 2007, 19, 173–191). This approach has been commonly used in numerous papers (citation). However, as the reviewer commented, this approach is not very precise, especially when there is no strict linear region in the absorption edge or when the light scattering is very significant for the absorption tail, which is the often case for the spin-coated organic films. Furthermore, open-shell radical molecules always show large absorption tail and lowest-energy absorption band originated from the presence of low-lying singlet excited state dominated by a doubly excited electronic configuration (H,H→L,L). (*J. Phys. Chem. Lett.* 2010, 1, 3334–3339) Therefore, the determination of absorbance onset is very subjective. To further give a declaration of the optical energy gap, we measured the absorption and the fluorescence emission spectra (Figure R1 and Figure S3) of all the samples, and presented the normalized curves together on the same abscissa axis. (*Adv. Energy Mater.* 2018, 8, 1801352) The optical bandgap is experimentally estimated as the overlap of absorption and fluorescence (zero-phonon line). The corresponding vertical transition from the initial ground state (lowest excited state) to most probable excited state (ground state) forms the maximum absorption $E_{\max,abs}$ (luminescence $E_{\max,pl}$). The relaxation occurs to excited state (ground state) form the relaxed energy λ_{abs} (luminescence λ_{fl}). The corresponding transition energy was defined as:

$$E_{\max,abs}=E_{0-0} + \lambda_{abs}$$

$$E_{\max,fl}=E_{0-0} - \lambda_{fl}$$

in which E_{0-0} is defined as the optical gap of the material. When the absorption and emission curves in the overlapped range are nearly symmetric, E_{0-0} (or E_g) can be defined as the energy at the intersection of normalized absorbance and emission spectra. This method to determine E_g are reproducible and physically credible. The updated data was summarized in Table R1. (Figure 2, Figure 3c, Table S1 in revised manuscript and supplementary information)

Figure R1. The photoluminescence spectra of (e) DPP-based small molecules with fused-phenyl groups, (f) TDPP, Th-TDPP, Flu-TDPP, TPAOMe-TDPP and TPAOMe-TDPP, (g) NT-based small molecules, and (h) TPAOMe-based small molecules in film.

Table R1. Optical and calculated electronic properties of the materials.

Materials	$\lambda_{\text{max}}^{\text{abs}}[\text{a}]$	$\lambda_{\text{max}}^{\text{pl}}[\text{b}]$	$\lambda_{\text{inter}}[\text{c}]$ [nm]	$E_g^{\text{opt}}[\text{d}]$ [eV]	HOMO ^e	LUMO ^f	E_g^{g} [eV]	y_0^{h}	y_1^{i}
Ph-TDPP	633	712	662	1.87	-4.81	-2.58	2.23	0.295	0.048
1N-TDPP	600	671	634	1.96	-4.82	-2.55	2.27	0.283	0.072
2N-TDPP	650	704	676	1.84	-4.78	-2.60	2.18	0.313	0.078
An-TDPP	582	642	614	2.02	-4.91	-2.48	2.43	0.237	0.156
Py-TDPP	630	698	668	1.86	-4.78	-2.59	2.19	0.310	0.142
TDPP	565	679	592	2.09	-4.97	-2.52	2.45	0.223	0.019
Th-TDPP	639	700	666	1.86	-4.78	-2.68	2.10	0.335	0.067
Flu-TDPP	639	740	670	1.85	-4.71	-2.56	2.15	0.317	0.073
TPAOMe-TDPP	660	730	694	1.79	-4.35	-2.29	2.06	0.309	0.056
TPAOMe-TDPP	693	978	784	1.58	-4.40	-2.50	1.90	0.379	0.112
NTT	543	623	603	2.06	-5.36	-2.88	2.48	0.269	0.054
2N-NTT	562	692	626	1.98	-5.12	-2.87	2.25	0.306	0.084
Flu-NTT	554	690	610	2.03	-5.04	-2.83	2.21	0.306	0.080
NTC	623	735	685	1.81	-4.80	-2.83	1.97	0.357	0.105
2N-NTC	652	817	738	1.68	-4.70	-2.86	1.84	0.396	0.160

Flu-NTC	672	909	778	1.59	-4.64	-2.83	1.81	0.396	0.159
TPAOMe-BTT	541	655	591	2.10	-4.47	-2.37	2.10	0.264	0.047
TPAOMe-NTT	574	725	646	1.92	-4.55	-2.68	1.87	0.304	0.071
TPAOMe-BBTT	879	/	1133 ^j	1.09	-4.29	-3.16	1.13	0.635	0.046

^aWavelength of maximum absorption from 300 to 1200 nm, ^bwavelength of maximum emission, ^ccalculated from the intersection of absorption and emission curves as pristine thin films, ^d $E_g^{opt}=1240/\lambda_{inter}$, ^eThe highest occupied molecular orbital (HOMO) and the ^flowest unoccupied molecular orbital (LUMO) energies calculated at B3LYP/6-31G(*d, p*) level theory and basis set. ^gThe calculated energy gap (E_g) between the HOMO and LUMO, ^hdiradical character index (y_0) and ⁱtetraradical character index (y_1) calculated with PUHF/6-31G(*d, p*). All energies are in eV, y_0 and y_1 are unitless quantities. ^jThe absorption wavelength of TPAOMe-BBTT was determined as the onset of the absorption curve because of the weak fluorescence.

Comment 2: Similarly the authors comment on the long absorption tail being characteristic of diradical character, but given that the measurements are not correctly for reflection and scattering, I don't think they can assert this. I strongly suggest they provide film measurements which have been corrected for these factors.

Response: We have defined the optical gaps as the intersection of normalized absorbance and emission curves. It can be more precise and therefore can eliminate the influence of reflection and scattering. Please see Table R1.

Comment 3: Why is all EPR data in the solid state only? What happens in solution measurements? Is diradical character observed? If not, why not. This should be addressed in the manuscript.

Response: Solution EPR spectra are very sensitive to molecular motion. Some conditions such as solvent viscosity, temperature, concentration can have a profound influence on the EPR spectra given their influence on molecular dynamics. (*Chem. Soc. Rev.*, 2018, 47, 2534-2553) Therefore, it was complicated to give a comparison between the solution samples. We have also complemented the EPR spectra of solution samples. It can be observed that the EPR response of the materials greatly weakened from powders to solutions (see the EPR spectra of 2N-NTC, NTC, TPAOMe-TDPP, TPAOMe-TTDP in powders and in toluene solutions, Figure R1). Some materials with weak solid-EPR response exhibited nearly invisible signal in solutions.

Figure R1. ESR spectra of (a) TPAOMe-TDPP, (b) TPAOMe-TTDPP, (c) NTC, (d) 2N-NTC in toluene solutions and in powders. The EPR spectra in powders and saturated solutions were conducted under the same test conditions in EPR sample tube.

We summarized the reasons of the greatly reduced solution-EPR intensity and are listed as followed.

- (1) The solvent will absorb the emitted microwave energy, which leading to a significant reduction of EPR signal in solution state. This phenomenon is more severe in polar solvents such as chloroform, chlorobenzene, and tetrahydrofuran (*Chem. Soc. Rev.*, 2005, 34, 164–178). Toluene is a relatively low polarity solvent in this measurement.
- (2) The dispersion of the solute molecules in solution results in a reduction of the overall spin density. It can be demonstrated by the similar phenomenon of the typical diradicaloid zethrene, which almost exhibited a nearly silent EPR response in solution. (*J. Am. Chem. Soc.* 2012, 134, 14913–14922) However, zethrene is commonly recognized as the analogue of polycyclic aromatic hydrocarbon embedded with *p*-QDM units, the diradical character of which have been proved by the experimental and theoretical experiments. (*J. Am. Chem. Soc.* 2012, 134, 14913–14922; *Acc. Chem. Res.* 2014, 47, 2582–2591)
- (3) The solution samples are saturated to ensure better reflection of intrinsic radical character and reduce the influence of solvent. However, the samples show different solubility in toluene due to the difference in molecular conformation and attached alkyl chains. For example, the concentration of TPAOMe-TDPP solution is 20 mg/ml, while the concentration of 2N-TDPP solution is 5 mg/ml due to the limited solubility. The intrinsic radical properties between these two analogues are hard to be presented.

Comment 4: Given the important of purity in these sensitive measurements, I find it surprising that no information on molecular purity is given except for NMR. Many of the mass spectra show peaks at different mass to that of the molecular ion. Given that all of these materials were made by Pd coupling, can the presence of homo-coupled defects or varying Pd levels be ruled out? Some information on purity (for example HPLC trace, melting points, elemental analysis) should be provided.

Response: We thank the reviewer for raising this concern. We provided more methods and spectra information for purity analysis. We conducted HPLC analysis with Cosmosil 5C₁₈-MS-II column. A single sharp peak can be observed for each molecule (as shown in Figure R2), which demonstrated the purity for these molecules. According to the DSC analysis, we detected a sharp and strong endothermic peak in the heating stage for some samples. It is a typical fingerprint characteristic assigned to the melting points. We mention that some samples such as Flu-NTC did not exhibit any endothermic peaks due to the weak crystallinity. Accordingly, we did not detect a phase transition from melting point determination. It may be due to the increase of flexible components including the multiple long alkyl chains (-C₁₂H₂₅) and the dimethyls on fluorenyls. The melting points of the materials are summarized in Table R2. Furthermore, the element composition (C, H, O, S, N) of the carefully purified and dried samples obtained through elemental analysis are nearly consistent with the calculated data (Table R3), further giving an evidence of the purity information. We are sorry for the unsatisfied mass spectra in the manuscript. The updated spectra are shown in Figure R3-R18. The amount of metallic impurities of the samples are measured by inductively coupled plasma mass spectrometry (ICP-MS) and particle-induced X-ray emission (PIXE). We found that the metallic impurities are lowered than 20 ppm. This amount of impurities is far from resulting the observed phenomenon in this manuscript. Therefore, we can eliminate the influence of Pd impurities.

Figure R2. HPLC spectra of (a, b) DPP-based, (c) NT-based, and (d) TPAOMe-based materials.

Table R2. Melting points of the materials.

Materials	Melting points (°C)	Materials	Melting points (°C)
NTT	182	Py-TDPP	217
2N-NTT	186	TDPP	/
Flu-NTT	217	Th-TDPP	186
NTC	128	Flu-TDPP	216
2N-NTC	174	TPAOMe-TDPP	207
Flu-NTC	/	TPAOMe-TDPP	240

Ph-TDPP	213	TPAOMe-BBT	/
1N-TDPP	183	TPAOMe-NTT	216
2N-TDPP	227	TPAOMe-BBTT	/
An-TDPP	209		

Table R3. Element components of the materials.

Materials	C(%) ^a	H(%) ^a	S(%) ^a	N(%) ^a	O(%) ^a
Ph-TDPP	74.52, 74.82	7.15, 7.12	9.47, 9.31	4.14, 4.20	4.73, 4.55
1N-TDPP	77.28, 77.21	6.75, 6.31	8.25, 8.11	3.60, 3.55	4.12, 4.82
2N-TDPP	77.28, 77.32	6.75, 6.33	8.25, 8.10	3.60, 3.50	4.12, 4.75
An-TDPP	79.41, 79.11	6.43, 6.41	7.31, 7.37	3.19, 3.15	3.65, 3.97
Py-TDPP	80.48, 80.15	6.10, 6.12	6.93, 7.06	3.03, 2.95	3.46, 3.73
Th-TDPP	66.24, 65.70	6.44, 6.22	18.61, 18.99	4.07, 3.95	4.64, 5.14
Flu-TDPP	79.25, 79.31	7.09, 7.13	7.05, 7.07	3.08, 3.10	3.52, 3.40
TPAOMe-TDPP	74.31, 73.95	6.59, 6.41	5.67, 5.71	4.95, 4.82	8.48, 9.11
TPAOMe-TDPP	72.30, 72.45	6.07, 6.17	9.90, 9.65	4.32, 4.26	7.41, 7.48
2N-NTT	72.43, 72.67	5.35, 5.29	15.47, 15.28	6.76, 6.67	/
Flu-NTT	74.96, 75.01	5.87, 5.92	13.34, 13.12	5.83, 5.81	/
NTC	71.87, 72.15	8.57, 8.41	15.15, 15.01	4.41, 4.67	/
2N-NTC	75.74, 76.11	7.95, 7.87	12.64, 12.26	3.68, 3.70	/
Flu-NTC	76.95, 77.10	8.04, 8.08	11.63, 11.29	3.39, 3.47	/
TPAOMe-BTT	74.30, 75.49	6.59, 5.15	8.50, 4.29	4.95, 7.47	5.66, 7.61
TPAOMe-NTT	71.04, 71.35	5.62, 5.55	10.83, 10.76	7.10, 7.01	5.41, 5.34
TPAOMe-BBTT	70.68, 71.21	6.10, 6.41	10.78, 10.40	7.06, 7.10	5.38, 4.88

^a Calcd, found

Comment 5: No info is given on how 2N-TDPP and 2N-NTT were sublimed.

Response: We are sorry for the missing of the detailed methods of sublimation. We have complemented the related descriptions in SI. The high vacuum sublimation was conducted at high vacuum of 0.01 Pa and temperatures above 300 °C for 72 hours. Nitrogen was injected into the tube after sublimation and subsequently, ESR tests were performed on sublimated solids. All the operations were performed in an inert environment and the isolation of metal iron (to eliminate the

influence of external ferromagnetism).

The replacement of SQUID data in Figure 4c and 4d

In the resubmitted manuscript, we have replaced the SQUID data in Figure 4c and 4d. The data in the initial submitted manuscript was wrongly collected because of the operational problems in measurement. The centering of the SQUID appears off, which results in a weak signal and relatively low singlet-triplet energy gaps. We carefully rerun the measurement to ensure the accuracy and authenticity of data.

Figure R3. The SQUID magnetometry of the solid sample showing magnetic susceptibility times temperature $\chi_M T$ vs. T from 200 – 400 K, fit to the Bleaney–Bowers equation with $g = 2.003$, giving ΔE_{ST} ($2J/k_B$) (red line).

The supplementary theoretical calculation data

We have analyzed the open-shell character of the materials with several methods to confirm their diradical and polyradical characters. The open-shell character of organic materials is strongly

dependent on the computational methods utilized (*Phys. Chem. Chem. Phys.* 2018, 20, 24227-38.). Using the spin-projected unrestricted Hartree-Fock (PUHF) method we find that all the materials possess variable open-shell character (Table S2), which correlates with their bandgap. As the PUHF method prone to a large spin contamination, we have tested the widely accepted broken symmetry (BS) approach with different density functionals. However, unlike the other open-shell small materials (*Nat. Chem.* 2016, 8, 753-9; *Nat. Chem.* 2018, 10, 1134-40) or polymers (*Mater. Adv.* 2021, 2, 2943–2955; *iScience.* 2020, 23, 101675.), the conventional BS approach did not predict open-shell character in the current set of molecules. For example, Rudebusch *et al.* predicted diradical character ($y_0 = 0.088$ to 0.273) for benzothiophene-based acenes using tuned LC-RBLYP CASCI(2,2) (Complete Active Space Configuration Interaction) (*Nat. Chem.* 2016, 8, 753-9). With the same approach, 2N-NTC and TPAOMe-TTDPP molecules provide a negligible diradical character ($y_0 = 0.003$ and 0.008 , respectively) indicating diradical character is not as pronounced in the current set of molecules than the ones reported by Haley and coworkers (*Nat. Chem.* 2016, 8, 753-9). We believe this to be due to the presence of numerous heteroatoms in the current set of molecules.

Another simple but elegant method to estimate the open-shell character is proposed by Grimme *et al.*, which is the fractional occupation number weighted electron density (N_{FOD}) (*Angew. Chem. Int. Ed.* 2015, 54, 12308-13.). Fractional orbital density (FOD) is an extremely simple and cost-effective method based on smearing the electrons over the molecular orbitals using finite temperature DFT (FT-DFT). N_{FOD} accurately quantifies the static electron correlation and molecules with a delocalized FOD and a large N_{FOD} have multireference character. Table R5 shows the N_{FOD} values of the molecules studied in this work. A large N_{FOD} values reveal that the electrons are strongly correlated, and different materials provide different N_{FOD} values. Interestingly, the trends are qualitatively consistent with diradical index computed from PUHF (see Table R4), showing good linear correlation between N_{FOD} and y_0 .

To better understand the radical nature for these molecules, we explore their open-shell characters (y_i) using fractional orbital occupancy. Table R6 shows the y_0 , y_1 , y_2 and y_3 values for all the molecules computed using the FT-DFT at B3LYP/6-31G(d,p) level of theory and basis set. Our result reveals polyradical character in all the molecules.

The spatial distribution of unpaired electrons in these molecules are evaluated using FOD plots (see Fig. R4-R6). FOD plots show partially delocalized/localized electron density distribution along the molecular backbones, disclosing strongly correlated electrons.

We have also computed the vertical singlet-triplet energy gap (ΔE_{ST}) using FT-DFT with B3LYP/6-31G(d,p) and the results are presented in Table R5. It was shown that the ΔE_{ST} gap computed using FT-DFT is comparable to that of CASPT2 method (*Phys. Chem. Chem. Phys.* 2018, 20, 7112-24.). The computed ΔE_{ST} gap for 2N-NTC (11.03 kcal/mol) and TPAOMe-TTDPP (10.65 kcal/mol) is overestimated compared to the experimental gap, 4.76 and 5.52 kcal/mol, respectively. We believe this is due to the medium effects that we are not able to capture in the isolated molecule calculations. However, we find a good correlation between y_0 (and N_{FOD}) and ΔE_{ST} gap using FT-DFT method, a larger N_{FOD} value indicates a smaller ΔE_{ST} gap.

Table R4. Calculated electronic and optical properties of the molecules.

Materials	PUHF ^a		FT-DFT ^b		λ^c (nm)	Transition ^d	Contribution ^e (%)	HOMO ^f (eV)	LUMO ^g (eV)	E_g^h (eV)
	y_0	y_1	y_0	y_1						
Ph-TDPP	0.295	0.048	0.481	0.118	556.85	H → L	97.5	-4.81	-2.58	2.23
1N-TDPP	0.283	0.072	0.476	0.136	546.88	H → L	96.8	-4.82	-2.55	2.27
2N-TDPP	0.313	0.078	0.581	0.163	567.68	H → L	96.6	-4.78	-2.60	2.18
An-TDPP	0.237	0.156	0.443	0.218	514.57	H → L	97.7	-4.91	-2.48	2.43
Py-TDPP	0.310	0.142	0.495	0.205	563.49	H → L	92.6	-4.78	-2.59	2.19
TDPP	0.223	0.019	0.431	0.070	502.31	H → L	99.0	-4.97	-2.52	2.45
Th-TDPP	0.335	0.067	0.519	0.150	589.68	H → L	96.7	-4.78	-2.68	2.10
Flu-TDPP	0.317	0.073	0.511	0.155	576.86	H → L	95.9	-4.71	-2.56	2.15
TPAOMe-TDPP	0.309	0.056	0.538	0.148	593.29	H → L	89.1	-4.35	-2.29	2.06
TPAOMe-TDPP	0.379	0.112	0.539	0.182	639.63	H → L	84.5	-4.40	-2.50	1.90
NTT	0.269	0.054	0.391	0.116	500.04	H → L	96.9	-5.36	-2.88	2.48
2N-NTT	0.306	0.084	0.434	0.132	539.84	H → L	93.0	-5.12	-2.87	2.25
Flu-NTT	0.306	0.080	0.443	0.136	546.80	H → L	91.3	-5.04	-2.83	2.21
NTC	0.357	0.105	0.466	0.145	616.24	H → L	94.3	-4.80	-2.83	1.97
2N-NTC	0.396	0.160	0.493	0.189	651.90	H → L	91.1	-4.70	-2.86	1.84
Flu-NTC	0.396	0.159	0.500	0.194	658.24	H → L	90.2	-4.64	-2.83	1.81
TPAOMe-BTT	0.264	0.047	0.445	0.102	536.80	H → L	75.7	-4.47	-2.37	2.10
TPAOMe-NTT	0.304	0.071	0.480	0.145	570.34	H → L	71.3	-4.55	-2.68	1.87
TPAOMe-BBT	0.635	0.046	0.805	0.093	1114.48	H → L	93.1	-4.29	-3.16	1.13

^aDiradical character index (y_0) and tetraradical character index (y_1) calculated with PUHF/6-31G(d,p).

^bDiradical character index (y_0) and tetraradical character index (y_1) calculated with FT-DFT/B3LYP/6-31G(d,p).

^cWavelength of the excitation from the ground to the first excited state. ^dOrbitals involved in the transition.

^eContribution of individual orbitals in the transition. ^fThe highest occupied molecular orbital (HOMO) and the ^glowest unoccupied molecular orbital (LUMO) energies calculated at RB3LYP/6-31G(d,p) level theory and basis set.

^hThe calculated energy gap (E_g) between the HOMO and LUMO. The excited state calculations are performed on the ground state geometry with PCM(chloroform)/TDDFT/BHandHLYP/6-31G(d,p) level of theory and basis set. All energies are in eV, λ is in nm. H = HOMO, L = LUMO

Table R5. N_{FOD} and vertical singlet-triplet energy gap for the molecules computed using FT-DFT at B3LYP/6-31G(d,p) level.

Materials	N_{FOD}	Vertical ΔE_{ST} gap (kcal/mol)
TDPP	1.104	23.02
1N-TDPP	1.801	15.47
2N-TDPP	2.081	13.64
An-TDPP	2.261	12.21
Th-TDPP	1.653	16.82
Flu-TDPP	1.946	14.75
Ph-TDPP	1.517	18.13
Py-TDPP	2.362	12.14
TPAOMe-TDPP	2.470	11.92
TPAOMe-TDPP	2.816	10.65
NTT	1.311	20.72
2N-NTT	2.016	14.40
Flu-NTT	2.093	14.07
NTC	1.909	14.95
2N-NTC	2.689	11.03
Flu-NTC	2.769	10.84
TPAOMe-BTT	2.277	13.08
TPAOMe-NTT	2.636	11.40
TPAOMe-BTT	2.968	10.20

Table R6. Computed radical indices (y_i) using FT-DFT at B3LYP/6-31G(d,p) level.

Materials	Fractional Orbital Occupancy			
	y_0	y_1	y_2	y_3
TDPP	0.431	0.070	0.018	0.009
1N-TDPP	0.476	0.136	0.095	0.055
2N-TDPP	0.581	0.163	0.094	0.070
An-TDPP	0.443	0.218	0.218	0.074
Th-TDPP	0.519	0.150	0.063	0.026
Flu-TDPP	0.511	0.155	0.084	0.043
Ph-TDPP	0.481	0.118	0.047	0.026
Py-TDPP	0.495	0.205	0.161	0.072
TPAOMe-TDPP	0.538	0.148	0.069	0.069
TPAOMe-TDPP	0.539	0.182	0.108	0.069
NTT	0.391	0.116	0.066	0.028
2N-NTT	0.434	0.133	0.112	0.095
Flu-NTT	0.443	0.136	0.115	0.094
NTC	0.466	0.145	0.121	0.088
2N-NTC	0.493	0.189	0.170	0.121
Flu-NTC	0.500	0.194	0.174	0.123
TPAOMe-BTT	0.445	0.102	0.089	0.068
TPAOMe-NTT	0.480	0.145	0.112	0.083
TPAOMe-BBTT	0.805	0.093	0.075	0.065

Figure R4. FOD plots ($\sigma = 0.002 \text{ e/Bohr}^3$) for DPP-based materials obtained from the FT-DFT at B3LYP/6-31G(d,p) level.

Figure R5. FOD plots ($\sigma = 0.002 \text{ e/Bohr}^3$) for NT-based materials obtained from the FT-DFT at B3LYP/6-31G(d,p) level.

Figure R6. FOD plots ($\sigma = 0.002 \text{ e/Bohr}^3$) for TPA-based molecules obtained from the FT-DFT at B3LYP/6-31G(d,p) level.

Figure R7. MALDI-TOF-MS of **Ph-TDPP**. Calcd. for $C_{42}H_{48}N_2O_2S_2$; m/z : 676.3157. Found: 676.3426.

Figure R8. MALDI-TOF-MS of **1N-TDPP**. Calcd. for $C_{50}H_{52}N_2O_2S_2$; m/z : 776.3470. Found: 776.3426.

Figure R9. MALDI-TOF-MS of **2N-TDPP**. Calcd for $C_{50}H_{52}N_2O_2S_2$: m/z : 776.3470. Found: 776.3467.

Figure R10. MALDI-TOF-MS of **An-TDPP**. Calcd for $C_{50}H_{52}N_2O_2S_2$: m/z : 876.3783. Found: 876.3773.

Figure R11. MALDI-TOF-MS of Py-TDPP. Calcd. for $C_{62}H_{56}N_2O_2S_2$; m/z : 924.3873. Found: 924.3883.

Figure R12. MALDI-TOF-MS of Flu-TDPP. Calcd for $C_{60}H_{64}N_2O_2S_2$; m/z : 908.4409. Found: 908.4425.

Figure R13. MALDI-TOF-MS of Flu-TDPP-C8. Calcd for $C_{60}H_{64}N_2O_2S_2$: m/z : 908.4409. Found: 908.4469.

Figure R14. MALDI-TOF-MS of TPAOMe-TDPP. Calcd for $C_{70}H_{74}N_4O_6S_2$: m/z : 1130.5050. Found: 1130.5164.

Figure R15. MALDI-TOF-MS of TPAOMe-TDPP-C4. Calcd for $C_{62}H_{58}N_4O_6S_2$: m/z : 1018.3798. Found:1018.3806.

Figure R16. MALDI-TOF-MS of TPAOMe-TDPP. Calcd for $C_{78}H_{78}N_4O_6S_4$: m/z : 1294.4804. Found: 1294.4776.

Figure R17. MALDI-TOF-MS of TPAOMe-BBTT. Calcd for $C_{78}H_{78}N_4O_6S_4$; m/z : 1188.4498. Found: 1188.4316.

Figure R18. MALDI-TOF-MS of 2N-NTT. Calcd for $C_{50}H_{44}N_4O_4S_4$; m/z : 828.2449. Found: 828.2546.

Figure R19. MALDI-TOF-MS of Flu-NTT. Calcd for $C_{60}H_{56}N_4S_4$; m/z : 960.3397. Found: 960.3388.

Figure R20. MALDI-TOF-MS of NTC. Calcd for $C_{76}H_{108}N_4S_6$; m/z : 1268.6898. Found: 1268.6955.

Figure R21. MALDI-TOF-MS of **2N-NTC**. Calcd for $C_{96}H_{120}N_4S_6$; m/z : 1521.7871. Found: 1521.7827.

Figure R22. MALDI-TOF-MS of **Flu-NTC**. Calcd for $C_{106}H_{132}N_4S_6$; m/z : 1653.8810. Found: 1653.8831.

REVIEWERS' COMMENTS

Reviewer #1 (Remarks to the Author):

This paper deals with electronic structure in open-shell donor-acceptor organic semiconductors. The NMR, ESR, Magnetic susceptibility measurements, single-crystal X-ray studies and computational investigation were carried out for the open-shell donor-acceptor organic semiconductors.

However, the results and discussion was not impressive. The design of donor-acceptor organic semiconductor was already carried out for the proper ICT to apply the proper application. The only interesting point is planar conjugated structure has ESR spectra. But the result was also already reported as described by author. In addition, this paper did not show any electronic application by using this phenomena. Therefore, I do not recommend this paper to publish in nature communications.

Reviewer #2 (Remarks to the Author):

The authors have addressed many of the concerns raised, and I believe the manuscript is indeed suitable for this journal. In particular the additional data regarding band gap and purity helps to partially eliminate that an impurity may be the cause of the effect, but given the sensitivity of EPR this is still a concern. One important issue still not addressed is the spin density in the sample. Whilst I don't expect that this be measured for every sample, for those materials showing the strongest apparent EPR signals it is important that the number be quantified. Is it significantly less than 1% per molecule, or higher? The spin density is important to further exclude impurities (which might not show up on HPLC or NMR, where the limit of detection is higher than for PER). This is relatively simple to measure versus a spin standard, and I think it is important to include. I would support publication once included.

Minor points:

This sentence does not make sense to me: 'Occasionally, in 2018 we found that Professor 88 Fred Wudl has also reported the similar results in narrow bandgap D-A molecules and copolymers based 89 on benzo[1,2-c;4,5-c]bis[1,2,5]thiadiazole (BBT) in 2015 (Fig. 1c) and we apologized that we ignored 90 this work and didn't cite it.31,32'

It reads like they apologised in reference 31 or 32; but I think they mean they apologize here? It shouldn't start with 'occasionally' either. Needs to be rephrased.

List of point-to-point response of reviewers' comments

Reviewer 1 (Remarks to the Author):

This paper deals with electronic structure in open-shell donor-acceptor organic semiconductors. The NMR, ESR, Magnetic susceptibility measurements, single-crystal X-ray studies and computational investigation were carried out for the open-shell donor-acceptor organic semiconductors.

However, the results and discussion was not impressive. The design of donor-acceptor organic semiconductor was already carried out for the proper ICT to apply the proper application. The only interesting point is planar conjugated structure has ESR spectra. But the result was also already reported as described by author. In addition, this paper did not show any electronic application by using this phenomena. Therefore, I do not recommend this paper to publish in nature communications.

Response: Thank you very much for your careful review and comments on the manuscript.

1) For the novelty of this work:

As noted by the reviewer, the interesting point is the planar conjugated structure has ESR spectra and the result was also already reported as described by us in the reference 31 (Yuan Li, et al., *J. Phys. Chem. C.* 2017, 121, 8579–8588). In fact, in our previous work, we reported and proposed that the intrinsic diradical character is widespread in the D-A type organic semiconductors. However, we only provided limited evidence to investigate open-shell singlet ground state of the D-A organic semiconductors.

Furthermore, we would like to point out that many researchers have made numerous efforts to develop novel material systems and focused on the photo- and electron-excited states, however, the in-depth investigations of ground-state electronic structures are relatively ignored. In the past 20 years, the detected ESR or paramagnetic species

in D-A organic semiconductors were recognized as impurities, defects, oxygen traps, polarons, radical cation/anions or charge states. (*Appl. Phys. Lett.* 2008, 93, 093303; *Chem. Commun.* 2015, 51, 2239-2241; *Adv. Mater.* 2018, 30, e1705052; *J. Org. Chem.* 2018, 83, 3651-3656). It is an extremely challenging job to disclose the origin of the ESR signal with various precise experimental technologies and theoretical methods. The fundamental understanding of the open-shell diradical character of the D-A organic semiconductors will further promote the rational design of these materials.

In this work, we demonstrate the open-shell diradical character of D-A type narrow bandgap organic semiconductors, and articulate the structure-property-electronic topology relationships within the D-A materials. The combination of experimental data and theoretical approaches demonstrate:

- a) The highly systematical examples of D-A materials with tunable diradical character ($y_0 = 0.22 \rightarrow 0.64$) stabilized in an intrachain donor-acceptor structure.
- b) Articulate the clear connections between the bandgap, π -extension, structural, and electronic features with the degree of diradical character.
- c) The diradical resonance form is widespread in narrow bandgap conjugated molecules such as those commonly employed in the fabrication of high-performing organic electronic devices.
- d) Disclose the evolution of the electronic structure elucidating, design guidelines to achieve open-shell character in this ubiquitous class of materials.

2) For the electronic applications by using the open-shell diradical character:

Reviewer 1 mentioned that this paper did not show any electronic application by using this phenomenon. We sincerely appreciate for the important comment and suggestion on our manuscript. Narrow bandgap organic semiconductors with intrachain donor-acceptor (D-A) structure and proper intramolecular charge transfer (ICT) effect showed superior physicochemical properties and have been developed in various applications in these years (Yang Yang et al., *Chem. Rev.* 2015, 115, 12633-12665; Alan

J. Heeger et al., *Chem. Soc. Rev.*, 2016, 45, 4825-4846). It was reported that the open-shell diradical materials based on D-A structure showed electronic applications in high-conducting materials (Jason D. Azoulay et al., *Adv. Funct. Mater.* 2020, 1909805), supercapacitors with high energy density and long cycle life (Jason D. Azoulay et al., *Adv. Energy Mater.* 2019, 9, 1902806), infrared organic photodetectors with high detectivity (Jason D. Azoulay et al., *Sci. Adv.* 2021; 7: eabg2418), and as photothermal agents in therapy and solar-to-vapor generations (Yuan Li et al., *Dyes and Pigments*, 2021, 192, 109460; Kanyi Pu et al., *ACS Nano* 2016, 10, 4472–4481). Importantly, all the work mentioned above have cited our previous work (Yuan Li, et al., *J. Phys. Chem. C.* 2017, 121, 8579–8588) as these papers showed the application potential of the D-A organic semiconductor materials.

This work is distinct from previous work on pro-aromatic or quinoidal Kékulé-type molecular systems as these D-A materials show significant internal charge transfer character, remarkable photo- and thermal stability, and practical application value in various fields. Since the report of Chichibabin's hydrocarbon in 1907 (Tschitschibabin, A. E. *Chem. Ber.* 1907, 40, 1810–1819), synthesis and isolation of stable diradical molecules is always in urgent (Jishan Wu et al., *Acc. Chem. Res.* 2014, 47, 2582–2591; Jaume Veciana et al., *Chem. Soc. Rev.* 2012, 41, 303–349). In the past few decades, some diradicaloids with relatively limited stability have been successfully isolated into pure compounds benefited from efficient and rational synthetic routes (Yuan Li et al., *Chem.* 2021, 7, 288-332; Michael M. Haley et al., *Nat. Chem.* 2016, 8, 753–759; Juan Casado, *Top. Curr. Chem.* 2017, 375, 73). However, the half-lifetime of these open-shell diradicaloids is mostly shorter than a month in inert atmosphere, and will significantly lose diradical character under heating or exposure to oxygen and water, which cannot meet the demand of practical applications. For example, the tetrabenzochichibabin's hydrocarbon reported by Jishan Wu showed a short life period of 2 days (Jishan Wu et al., *J. Am. Chem. Soc.* 2012, 134, 14513-14525). Triplet-ground-state diradical based on diindenopyrazine exhibited a half-lifetime of 22 days (Jie-Yu Wang et al., *Angew. Chem. Int. Ed.* 2021, 60, 4594).

In contrast, D-A materials based on DPP and NT building blocks showed superior stability and were successfully applied in optoelectronic, thermoelectric and photothermal technologies. For the DPP-based materials:

- a) Organic photovoltaics. (Erjun Zhou et al., *Adv. Mater.* 2017, 29, 1600013; Weiwei Li et al. *Acc. Chem. Res.* 2016, 49, 78–85)
- b) Organic field-effect transistors. (Christian B. Nielsen et al., *Adv. Mater.* 2013, 25, 1859-1880; Yanhou Gen et al., *Adv. Sci.* 2019, 6, 1902412; Yanhou Gen et al., *Adv. Funct. Mater.* DOI: 10.1002/adfm.202104881).
- c) Candidate for efficient singlet fission. (Tobin J. Marks et al., *J. Am. Chem. Soc.* 2016, 138, 11749–11761; Satish Patil et al. *Nat Commun.* 2019, 10, 33)
- d) Photoacoustic imaging and photodynamic/photothermal therapy. (Wei Huang et al., *ACS Nano* 2017, 11, 1054–1063, Yuan Li et al., *Dyes and Pigments*, 2021, 192, 109460)
- e) Stretchable electronic applications. (Zhenan Bao et al., *Nat. Commun.* 2021, 12, 3572)
- f) Nonlinear optical materials. (Deqing Zhang et al., *ACS Appl. Mater. Interfaces* 2020, 12, 2, 2944–2951)
- g) Sensors. (Woo-Dong Jang et al., *Chem. Asian. J.* 7, 1562–1566; Andreas Zumbusch et al., *Chem. Commun.*, 2014,50, 4755-4758)

For the NT-based materials:

- a) Organic photovoltaics. (Fei Huang et al., *J. Am. Chem. Soc.* 2011, 133, 9638–9641; Fei Huang et al. 2016, *Adv. Mater.*, 28, 9811-9818; He yan et al.; *Nat. Commun.* 2014, 5, 5293)
- b) Organic photodetectors. (Fei Huang et al., *Nat Commun.* 2020, 11, 2871; Fei Huang et al., *J. Mater. Chem. C*, 2019,7, 6070-6076).
- c) Organic field-effect transistors. (Kazuo Takimiya et al., *Macromolecules* 2015, 48, 576–584; Yen-Ju Cheng et al., *J. Mater. Chem. C*, 2016, 4, 11427-11435)

In this work, we did not focus on applications as we did not have adequate evidence to clearly present underlying mechanism and structure-properties-performance relationship. We are trying the diradical compounds as charge transport layer in OFETs, emission layer in OLEDs, as singlet fission candidates in OPVs, photothermal and thermoelectric areas. These projects are currently under way in our laboratory.

Reviewer #2 (Remarks to the Author):

The authors have addressed many of the concerns raised, and I believe the manuscript is indeed suitable for this journal. In particular the additional data regarding band gap and purity helps to partially eliminate that an impurity may be the cause of the effect, but given the sensitivity of EPR this is still a concern. One important issue still not addressed is the spin density in the sample. Whilst I don't expect that this be measured for every sample, for those materials showing the strongest apparent EPR signals it is important that the number be quantified. Is it significantly less than 1% per molecule, or higher? The spin density is important to further exclude impurities (which might not show up on HPLC or NMR, where the limit of detection is higher than for PER). This is relatively simple to measure versus a spin standard, and I think it is important to include. I would support publication once included.

Response: Thank you so much for your affirmation and the insightful interpretations. We sincerely express our heartiest thanks to you because your key and important comments guided us to study our results more deeply and factually improve the manuscript quality. We will response to all the comments point by point carefully.

You raised the concern of spin density of the D-A compounds. The spin concentration correlated with the ESR intensity is an important concern for researchers in the field of organic radical materials. Using the monoradical DPPH ($S=1/2$) as the standard, the spin concentration (spin density) of TPAOMe-TDPP and 2N-NTC were carefully tested and calculated as 0.15 % and 0.17 % N_A , respectively (Fig. R1). This low triplet

spin concentration is expected as the variable temperature and susceptibility measurements confirmed the singlet ground state electronic structure with large singlet-triplet energy splitting of these compounds (ΔE_{ST} of TPAOMe-TTDPP and 2N-NTC is -5.52 kcal/mol and -4.76 kcal/mol), indicating the weak contribution of paramagnetic triplet in ground state.

Fig. R1. ESR spectra of TPAOMe-TTDPP, 2N-NTC and standard monoradical DPPH. The signal intensity is amplified 50 times for visual representation. The ESR measurements were conducted under the same conditions and using the same molar quantity of each material at 0.01 mmol.

It must be noted that the well-known diradicaloid indenoindenodibenzothiophene exhibited silent ESR response at room temperature, in consistent with its large ΔE_{ST} of -8.8 kcal/mol (Michael M. Haley et al., *Nat. Chem.* 2018, 10, 1134–1140). The monoradical DPPH showed weak spin-spin interaction and strong paramagnetic property. When an external magnetic field (H) is applied in a direction, the electron's magnetic moment aligns itself either parallel ($m_s = -1/2$, β spin) or antiparallel ($m_s = +1/2$, α spin) to the field (Fig. R2), contributing to its high ESR intensity. Diradicals (bipolarons in Fig. R3) showed a weakened ESR signal because of the intramolecular and intermolecular interaction.

Fig. R2. Zeeman effect on unpaired electrons and the ESR spectrum for monoradical system. Electron spin resonance (ESR) occurs when the frequency (ν) is adjusted to the energy of $\Delta E = E_\alpha - E_\beta = g_e \mu_e H = h\nu$. (Manabu Abe, *Chem. Rev.* 2013, 113, 7011–7088)

Fig. R3. Schematic diagrams illustrating the cases of a confined π -system (shown as a box) where carriers can interact to give strong interchain interactions (top) and the case wherein a system is isolated so as to be properly described as an intrachain carrier (bottom). (Timothy M. Swager, *Macromolecules*, 2017, 50, 4867–4886)

To study the spin density and confirm the intrinsic diradical character of the D-A-D compounds in our manuscript, we synthesized a well-known stable *p*-quinodimethane

(*p*-QDM) molecule CN-TDPP with open-shell singlet ground state and compared the ESR intensity with that of TPAOMe-TDPP under the same test condition. Since the first report of Chichibabin's hydrocarbon in 1907, investigation on the diradical molecules based on quinoidal *p*-QDMs have attracted wide concerns. These molecules showed a resonant conversion from quinoidal to open-shell diradical form (Fig. R4). The recovery of aromaticity in the quinoidal rings compensates the energy of the facture of double bonds, helping to stabilize the diradical structure (Manabu Abe, *Chem. Rev.* 2013, 113, 7011–7088), which can be interpreted as the Clar's aromatic sextet rule. In the past over 110 years, multiple diradical systems including the quinoidal oligothiophenes and quinoidal diketopyrrolopyrrole derivatives have been reported following the Chichibabin's hydrocarbon and other similar design concept (Fig. R4).

Fig. R4. Resonance structures of Chichibabin's hydrocarbon, quinoidal oligothiophenes, and quinoidal diketopyrrolopyrrole derivatives. (Tschitschibabin, A. E., *Chem. Ber.* 1907, 40, 1810–1819; Juan Casado et al., *J. Am. Chem. Soc.* 2002, 124, 12380-12388; Xiaozhang Zhu et al., *Angew. Chem. Int. Ed.* 2019, 58, 11291-11295; Daoben Zhu et al., *J. Am. Chem. Soc.* 2012, 134, 4084–4087; Satish Patil et al., *J. Phys. Chem. C* 2017, 121, 16088–16097)

Fig. R5. (a) Resonance structures of Chichibabin's analogue CN-TDPP and TPAOMe-TTDPP in this work. (b) ESR spectra of the two small molecules. The measurements were conducted under the same conditions and using the same molar quantity of each material at 0.02 mmol.

CN-TDPP is one of the typical diradicaloid belonging to the quinoidal *p*-QDMs family (Fig. R5). Interestingly, these two molecules showed ESR signals of very similar intensity. The spin concentration of CN-TDPP and TPAOMe-TTDPP was 0.18 % N_A and 0.15 % N_A , respectively. The spin concentrations of these two compounds are in a similar order, indicating that the diradical character of the D-A-D type compound TPAOMe-TTDPP is reasonable and acceptable.

To further demonstrate the rationality of low spin concentration, we complemented some discussions of the Chichibabin's diradical molecules. Chichibabin's hydrocarbon and its analogues were reported in the beginning of last century, and were widely recognized having open-shell diradical character in their ground state (Fig. R6). However, the intensity of the ESR intensity of these typical diradical molecules are not always very strong depending on the interaction and coupling effect of the two unpaired radicals. Diradical materials with distorted conformation exhibited weak intramolecular spin-spin interaction because of the remarkable steric protection effect. In this case, the diradical is more likely to behave as two monoradicals, thus significantly enhancing the ESR intensity. For example, Jishan Wu et al. benzannulated the central biphenyl unit of Chichibabin's hydrocarbon with four aromatic benzene rings, providing tetrabenzo-

Chichibabin's hydrocarbons with distorted conformational structure according to the single-crystal structure (Fig. R6) (Jishan Wu, *J. Am. Chem. Soc.* 2012, 134, 14513–14525). The significant intra- and intermolecular steric effect of the benzannulated Chichibabin's hydrocarbon by blocking the reactive sites with two bulky anthracene unit ensures the better air stability and higher diradical character comparing with Chichibabin's hydrocarbon. The other example is the extended tetracyano-oligo(*N*-annulated perylene)quinodimethanes (nPer-CN with $n = 1-6$) (Jishan Wu et al., *J. Am. Chem. Soc.* 2013, 135, 6363–6371). The shortest member of the series, 1Per-CN, can be identified by having a closed-shell electronic structure with a planar molecular conformation. As the conjugation enlarges, the conformational distortion greatly weakens the interaction of terminal spins. The longer oligomers 5Per-CN and 6Per-CN exhibited the open-shell triplet ground state, as demonstrated by the experimental and theoretical evidence.

Fig. R6. (a) Resonant structures of Chichibabin's hydrocarbon and its benzannulated analogue. (b) Single-crystal structure of the benzannulated Chichibabin's hydrocarbon. (Jishan Wu et al., *J. Am. Chem. Soc.* 2012, 134, 14513–14525) (c, d) Resonant structures and ESR spectra of the extended tetracyano-oligo(*N*-annulated perylene)quinodimethanes (nPer-CN with $n = 1-6$) (Jishan Wu et al., *J. Am. Chem. Soc.* 2013, 135, 6363–6371).

Diradicals and diradicaloids with planar molecular structure tended to exhibit weak ESR intensity because that the two radicals feature a strong intra- and intermolecular covalent interaction with antiparallel pairing. For example, octazethrene (Fig. R7a, Jishan Wu et al., *J. Am. Chem. Soc.*, 2012, 134, 14913–14922), a typical polycyclic aromatic hydrocarbon with open-shell singlet ground state, showed a relatively weak ESR response in solid state, however, it was ESR-silent in solution. According to the single crystal X-ray diffraction studies, octazethrene possessed a planar molecular conformation (Fig. R7a). The strong spin-spin interaction within the planar structure results in the weak ESR intensity. Michael M. Haley reported the indenoindenodibenzothiophene diradical molecule (Fig. R7b), which is unique for its low-energy-lying thermal triplet state and moderately strong diradical character (Michael M. Haley et al., *Nat. Chem.* 2018, 10, 1134–1140). This molecule is ESR-silent at room temperature because of the large contribution of singlet diradical in ground state.

Fig. R7. Resonance structure of octazethrene and indenoindenodibenzothiophene diradical, and their single crystal structures. (Jishan Wu et al., *J. Am. Chem. Soc.*, 2012, 134, 14913–14922; Michael M. Haley et al., *Nat. Chem.* 2018, 10, 1134–1140)

Besides, in the valence bond description, the spin-spin interaction is not only represented by the resonance within a molecule, but also by the intermolecular

interaction, as shown schematically in Fig. R8. Intermolecular interaction between the spins will greatly affect the spin concentration. Materials with planar molecular conformation and small π - π distance feature a strong intermolecular covalent character, and thus exhibited weak ESR intensity (Fig. R8).

Fig. R8. Resonance structures of intra- and intermolecular interactions of two unpaired electrons in the 1D chain. Wavy lines denote electron-electron interactions. Picture from Takashi Kubo et al. *Angew. Chem. Int. Ed.* 2005, 44, 6564-6568.

Minor points:

This sentence does not make sense to me: ‘Occasionally, in 2018 we found that Professor 88 Fred Wudl has also reported the similar results in narrow bandgap D-A molecules and copolymers based 89 on benzo[1,2-c;4,5-c]bis[1,2,5]thiadiazole (BBT) in 2015 (Fig. 1c) and we apologized that we ignored 90 this work and didn’t cite it.31,32’ It reads like they apologised in reference 31 or 32; but I think they mean they apologize here? It shouldn’t start with ‘occasionally’ either. Needs to be rephrased.

Response: We are sorry for the mistake. Thank you very much for your comment. We have changed this sentence as follow.

After we published our work, we found that Prof. Wudl *et al.* has reported the open-

shell character of the narrow bandgap D-A small molecules and polymers based on benzo[1,2-*c*;4,5-*c'*]bis[1,2,5]thiadiazole (BBT) in 2015 (Fig. 1c),³² and we regret this oversight on our part.³¹